# FIRST-ORDER OPTIMIZATION INSPIRED FROM FINITE-TIME CONVERGENT FLOWS

## ABSTRACT

In this paper, we investigate the performance of two first-order optimization algorithms, obtained from forward Euler discretization of finite-time optimization flows. These flows are the rescaled-gradient flow (RGF) and the signed-gradient flow (SGF), and consist of non-Lipscthiz or discontinuous dynamical systems that converge locally in finite time to the minima of gradient-dominated functions. We propose an Euler discretization for these first-order finite-time flows, and provide convergence guarantees, in the deterministic and the stochastic setting. We then apply the proposed algorithms to academic examples, as well as deep neural networks training, where we empirically test their performances on the SVHN dataset. Our results show that our schemes demonstrate faster convergences against standard optimization alternatives.

## 1 INTRODUCTION

Consider the unconstrained minimization problem for a given cost function $f : \mathbb{R}^n \to \mathbb{R}$. When $f$ is sufficiently regular, the standard algorithm in continuous time (dynamical system) is given by

$$\dot{x} = F_{\text{GF}}(x) \triangleq -\nabla f(x) \tag{1}$$

with $\dot{x} \triangleq \frac{\mathrm{d}}{\mathrm{d}t} x(t)$, known as the *gradient flow* (GF). Generalizing GF, the *q-rescaled* GF (q-RGF) Wibisono et al. (2016) given by

$$\dot{x} = F_{q-\text{RGF}}(x) = -c \frac{\nabla f(x)}{\|\nabla f(x)\|_2^{\frac{q-2}{q-1}}} \tag{2}$$

with $c > 0$ and $q \in (1, \infty]$ has an asymptotic convergence rate $f(x(t)) - f(x^\star) = \mathcal{O}\left(\frac{1}{t^{q-1}}\right)$ under mild regularity, for $\|x(0) - x^\star\| > 0$ small enough, where $x^\star \in \mathbb{R}^n$ denotes a local minimizer of $f$. However, it has been recently proven Romero & Benosman (2020) that q-RGF, as well as *q-signed* GF (q-SGF)

$$\dot{x} = F_{q-\text{SGF}}(x) = -c \|\nabla f(x)\|_1^{\frac{1}{q-1}} \text{sign}(\nabla f(x)), \tag{3}$$

where $\text{sign}(\cdot)$ denotes the sign function (element-wise), are both finite-time convergent (in continuous-time), provided that $f$ is gradient dominated of order $p \in (1, q)$. In particular, if $f$ is strongly convex, then q-RGF and q-SGF is finite-time convergent for any $q \in (2, \infty]$, since $f$ must be gradient dominated of order $p = 2$. Considering that many algorithms are inspired from continuous flows with convergence guarantee, e.g., Muehlebach & Jordan (2019); Fazlyab et al. (2017a); Shi et al. (2018); Zhang et al. (2018); França et al. (2019b); Wibisono et al. (2016), a natural question arises: *what is the convergence rate of the corresponding discrete-time algorithms, which are induced by discretization of finite-time convergent continuous-time flows?*

### 1.1 CONTRIBUTION

In this paper, we investigate the convergence behavior of an Euler discretization for the q-RGF (equation 2) and q-SGF (equation 3). We provide convergence guarantees in terms of closeness of solutions, using results from hybrid dynamical control theory. Furthermore, we provide iteration / sample complexity upper bounds for both the general (deterministic) and stochastic settings. We then test the performance of the proposed algorithms on both synthetic and real-world data in the context of deep learning, namely, on the well-known SVHN dataset.

## 1.2 RELATED WORK

Propelled by the work of Wang & Elia (2011) and Su et al. (2014), there has been a recent and significant research effort dedicated to analyzing optimization algorithms from the perspective of dynamical systems and control theory, especially in continuous time (Wibisono et al., 2016; Wilson, 2018; Lessard et al., 2016; Fazlyab et al., 2017b; Scieur et al., 2017; Franca et al., 2018; Fazlyab et al., 2018; Fazlyab et al., 2018; Taylor et al., 2018; França et al., 2019a; Orvieto & Lucchi, 2019; Romero et al., 2019; Muehlebach & Jordan, 2019). A major focus within this initiative is in *accceleration*, both in terms of trying to gain new insight into more traditional optimization algorithms from this perspective, or even to exploit the interplay between continuous-time systems and their potential discretizations for novel algorithm design (Muehlebach & Jordan, 2019; Fazlyab et al., 2017a; Shi et al., 2018; Zhang et al., 2018; França et al., 2019b; Wilson et al., 2019). Many of these papers also focus on deriving convergence rates based on the discretization of flows designed in the continuous-time domain.

Connecting ordinary differential equations (ODEs) and their numerical analysis, with optimization algorithms is a very important topic, which can be dated back to 1970s, see Botsaris (1978a;b); Zghier (1981); Snyman (1982; 1983); Brockett (1988); Brown (1989). In Helmke & Moore (1994), the authors studied relationships between linear programming, ODEs, and general matrix theory. Further, Schropp (1995) and Schropp & Singer (2000) explored several aspects linking nonlinear dynamical systems to gradient-based optimization, including nonlinear constraints.

Tools from Lyapunov stability theory are often employed for the analysis, mainly because there already exists a rich body of works within the nonlinear systems and control theory community for this purpose. In particular, typically in previous works, one seeks asymptotically Lyapunov stable gradient-based systems with an equilibrium (stationary point) at an isolated extremum of the given cost function, thus certifying local convergence. Naturally, the global asymptotic stability leads to global convergence, though such analysis will typically require the cost function to be strongly convex everywhere.

For physical systems, a Lyapunov function can often be constructed from first principles via some physically meaningful measure of energy (*e.g.*, total energy = potential energy + kinetic energy). In optimization, the situation is somewhat similar in the sense that a suitable Lyapunov function may often be constructed by taking simple surrogates of the objective function as candidates. For instance, $V(x) \triangleq f(x) - f(x^\star)$ can be a good initial candidate. Further, if $f$ is continuously differentiable and $x^\star$ is an isolated stationary point, then another alternative is $V(x) \triangleq \|\nabla f(x)\|^2$.

However, most fundamental and applied research conducted in systems and control regarding Lyapunov stability theory deals exclusively with *continuous-time* systems. Unfortunately, (dynamical) stability properties are generally not preserved for simple forward-Euler and sample-and-hold discretizations and control laws (Stuart & Humphries, 1998). Furthermore, practical implementations of optimization algorithms in modern digital computers demand discrete-time. Nonetheless, it has been extensively noted that a vast amount of general Lyapunov-based results (perhaps most) appear to have a discrete-time equivalent.

## 2 PRELIMINARIES

### 2.1 OPTIMIZATION ALGORITHMS AS DISCRETE-TIME SYSTEMS

Generalizing (1), (2), and (3), consider a continuous-time algorithm (dynamical system) modeled via an ordinary differential equation (ODE)

$$\dot{x} = F(x) \tag{4}$$

for $t \geq 0$, or, more generally, a differential inclusion

$$\dot{x}(t) \in \mathcal{F}(x(t)) \tag{5}$$

a.e. $t \geq 0$ (*e.g.* for the $q = \infty$ case), such that $x(t) \to x^\star$ as $t \to t^\star$. In the case of the $q$-RGF (2) and $q$-SGF (3) for $f$ gradient dominated of order $p \in (1, q)$, we have finite-time convergence, and thus $t^\star = t^\star(x(0)) < \infty$.

Most of the popular numerical optimization schemes can be written in a state-space form (*i.e.*, recursively), as

$$X_{k+1} = F_{\mathrm{d}}(k, X_k) \tag{6a}$$
$$x_k = G(X_k) \tag{6b}$$

for $k \in \mathbb{Z}_+ \triangleq \{0, 1, 2, \dots\}$ and a given $X_0 \in \mathbb{R}^m$ (typically $m \geq n$), where $F_{\mathrm{d}} : \mathbb{Z}_+ \times \mathbb{R}^m \to \mathbb{R}^m$ and $G : \mathbb{R}^m \to \mathbb{R}^n$.

Naturally, (6) can be seen as a discrete-time dynamical system constructed by discretizing (4) in time. In particular, we have $x_k \approx x(t_k)$, where $\{0 = t_0 < t_1 < t_2 < \dots\}$ denotes a time partition and $x(\cdot)$ a solution to (4) or (5) as appropriate. Therefore, we call $X_k$ and $x_k$, respectively, the *state* and *output* at time step $k$.

**Example 1.** The standard gradient descent (GD) algorithm

$$x_{k+1} = x_k - \eta \nabla f(x_k) \tag{7}$$

with step size (learning rate) $\eta > 0$ can be readily written in the form (6) by taking $m = n$, $F_{\mathrm{d}}(x) \triangleq x - \eta \nabla f(x)$, and $G(x) \triangleq x$.

- If the step sizes are adaptive, *i.e.* if we replace $\eta$ by a sequence $\{\eta_k\}$ with $\eta_k > 0$, then we only need to replace $F_{\mathrm{d}}(k, x) \triangleq x - \eta_k \nabla f(x)$, provided that $\{\eta_k\}$ is not computed using feedback from $\{x_k\}$ (*e.g.* through a line search method).

- If we do wish to use time-varying step-size, then we can set $m = n + 1$, $G([x; \eta]) \triangleq x$, and $F_{\mathrm{d}}([x; \eta]) \triangleq [F_{\mathrm{d}}^{(1)}([x; \eta]); F_{\mathrm{d}}^{(2)}([x; \eta])]$, where $F_{\mathrm{d}}^{(1)}([x; \eta]) \triangleq x - \eta \nabla f(x)$, and $F_{\mathrm{d}}^{(2)}$ is a user-defined function that dictates the updates in the step size. In particular, an open-loop adaptive step size $\{\eta_k\}$ may be achieved under this scenario, provided that it is possible to write $\eta_{k+1} = F_{\mathrm{d}}^{(2)}(\eta_k)$.

- If we wish to use individual step sizes for each the $n$ components of $\{x_k\}$, then it suffices to take $\eta_k$ as an $n$-dimensional vector (thus $m = 2n$), and make appropriate changes in $F_{\mathrm{d}}$ and $G$.

In each of these cases, GD can be seen as a forward-Euler discretization of the GF (1), *i.e.*,

$$x_{k+1} = x_k + \Delta t_k F_{\mathrm{GF}}(x_k) \tag{8}$$

with $F_{\mathrm{GF}} = -\nabla f$ and adaptive time step $\Delta t_k \triangleq t_{k+1} - t_k$ chosen as $\Delta t_k = \eta_k$.

**Example 2.** The proximal point algorithm (PPA)

$$x_{k+1} = \arg\min_{x \in \mathbb{R}^n} \left\{ f(x) + \frac{1}{2\eta_k} \|x - x_k\|_2^2 \right\} \tag{9}$$

with step size $\eta_k > 0$ (open loop, for simplicity) can also be written in the form (6), by taking $m = n$, $F_{\mathrm{d}}(k, x) \triangleq \arg\min_{x' \in \mathbb{R}^n} \{f(x') + \frac{1}{2\eta_k} \|x' - x\|_2^2\}$, and $G(x) \triangleq x$. Naturally, we need to assume sufficient regularity for $F_{\mathrm{d}}(k, x)$ to exist and we must design a consistent way to choose $F_{\mathrm{d}}(k, x)$ when multiple minimizers exist in the definition of $F_{\mathrm{d}}(k, x)$. Alternatively, these conditions must be satisfied, at the very least, at every $(k, x) \in \{(0, x_0), (1, x_1), (2, x_2), \dots\}$ for a particular chosen initial $x_0 \in \mathbb{R}^n$.

By assuming sufficient regularity, we have $\nabla_x \{f(x) + \frac{1}{2\eta_k} \|x - x_k\|_2^2\}|_{x=x_{k+1}} = 0$, and thus

$$\nabla f(x_{k+1}) + \frac{1}{\eta_k}(x_{k+1} - x_k) = 0 \iff x_{k+1} = x_k + \Delta t_k F_{\mathrm{GF}}(x_{k+1}) \tag{10}$$

with $\Delta t_k = \eta_k$, which is precisely the backward-Euler discretization of the GF (1).

## 2.2 CONTINUOUS-TIME CONVERGENCE OF $q$-RGF AND $q$-SGF

In this section, we review the necessary conditions to ensure finite-time convergence of these flows. Here the hyperparameter $c > 0$ in equation 2 and equation 3 will not be explicitly denoted in $F_{q-\mathrm{RGF}}, F_{q-\mathrm{SGF}}$. Next, borrowing terminologies from Wilson et al. (2019), we define the notion of gradient dominance with order $p$ as following.

**Assumption 1** (Gradient Dominance of Order $p$). For a continuously differentiable function $f$, we assume the function $f$ is $\mu$-*gradient dominated of order* $p \in (1, \infty]$ ($\mu > 0$), i.e.,

$$\frac{p-1}{p}\|\nabla f(x)\|_2^{\frac{p}{p-1}} \geq \mu^{\frac{1}{p-1}}(f(x) - f(x^\star)), \quad \forall\, x \in \mathbb{R}^n, \tag{11}$$

where $x^\star \in \arg\min_{x \in \mathbb{R}^n} f(x)$ is the local minimizer, also we denote the optimal value $f^\star \triangleq f(x^\star)$.

**Remark 1.** It can be proved that continuously differentiable strongly convex functions are gradient dominated of order $p = 2$. In fact, gradient dominance is usually defined exclusively for order $p = 2$, often referred to as the Polyak-Łojasiewicz (PL) inequality, which was introduced by Polyak (1963) to relax the (strong) convexity assumption commonly used to show convergence of the GD algorithm (7). The PL inequality can also be used to relax convexity assumptions of similar gradient and proximal-gradient methods (Karimi et al., 2016; Attouch & Bolte, 2009). Furthermore, if $f$ is gradient dominated (of any order) w.r.t. $x^\star$, then $x^\star$ is an isolated stationary point of $f$.

Our adopted generalized notion of gradient dominance is strongly tied to the Łojasiewicz gradient inequality from real analytic geometry, established by Łojasiewicz (1963; 1965)[1] independently and simultaneously from Polyak (1963), and generalizing the PL inequality. More precisely, this inequality is typically written as: for some $C > 0$ and $\theta \in \left(\frac{1}{2}, 1\right]$,

$$\|\nabla f(x)\|_2 \geq C \cdot |f(x) - f^\star|^\theta \tag{12}$$

holds for every $x \in \mathbb{R}^n$ in a small enough open neighborhood of the stationary point $x = x^\star$. This inequality is guaranteed for analytic functions Łojasiewicz (1965). More precisely, when $x^\star$ is a local minimizer of $f$, the aforementioned relationship is explicitly given by

$$C = \left(\frac{p}{p-1}\right)^{\frac{p-1}{p}} \mu^{\frac{1}{p}}, \qquad \theta = \frac{p-1}{p}. \tag{13}$$

Therefore, analytic functions are always gradient dominated. However, while analytic functions are always smooth, smoothness is not required to attain gradient dominance. Also recently it is shown that in reinforcement learning, the value functions with softmax parameterization satisfies the above condition (Mei et al., 2020; 2021), which further rationalizes our settings.

The following results in Romero & Benosman (2020) summarized the finite-time convergence of $q$-RGF (equation 2) and $q$-SGF (equation 3) in continuous-time sense, which also motivates the main topic in this paper.

**Theorem 1** (Romero & Benosman (2020)). *Suppose that $f : \mathbb{R}^n \to \mathbb{R}$ is continuously differentiable and $\mu$-gradient dominated of order $p \in (1, \infty)$ near a strict local minimizer $x^\star \in \mathbb{R}^n$. Let $c > 0$ and $q \in (p, \infty]$. Then, any maximal solution $x(\cdot)$, in the sense of Filippov, to the $q$-RGF (2) or $q$-SGF (3) will converge in finite time to $x^\star$, provided that $\|x(0) - x^\star\|_2 > 0$ is sufficiently small. More precisely, $\lim_{t \to t^\star} x(t) = x^\star$, where the convergence time $t^\star < \infty$ may depend on which flow is used, but in both cases is upper bounded by*

$$t^\star \leq \frac{\|\nabla f(x_0)\|_2^{\frac{1}{\theta} - \frac{1}{\theta'}}}{cC^{\frac{1}{\theta}}\left(1 - \frac{\theta}{\theta'}\right)}, \tag{14}$$

*where $x_0 = x(0)$, $C = \left(\frac{p}{p-1}\right)^{\frac{p-1}{p}}\mu^{\frac{1}{p}}$, $\theta = \frac{p-1}{p}$, and $\theta' = \frac{q-1}{q}$. In particular, given any compact and positively invariant subset $S \subset \mathcal{D}$, both flows converge in finite time with the aforementioned convergence time upper bound (which can be tightened by replacing $\overline{\mathcal{D}}$ with $S$) for any $x_0 \in S$. Furthermore, if $\mathcal{D} = \mathbb{R}^n$, then we have global finite-time convergence, i.e. finite-time convergence to any maximal solution (in the sense of Filippov) $x(\cdot)$ with arbitrary $x_0 \in \mathbb{R}^n$.*

In essence, the analysis introduced in Romero & Benosman (2020) consists of leveraging the gradient dominance to show that the energy function $\mathcal{E}(t) \triangleq f(x(t)) - f^\star$ satisfies the Lyapunov-like differential inequality $\dot{\mathcal{E}}(t) = \mathcal{O}(\mathcal{E}(t)^\alpha)$ for some $\alpha < 1$. The detailed proof is recalled in Appendix C for completeness.

---

[1]For more modern treatments in English, see Łojasiewicz & Zurro (1999); Bolte et al. (2007)

# 3 MAIN RESULTS: CONVERGENCE ANALYSIS FOR EULER DISCRETIZATION

With the previous discussion on continuous-time gradient flow, now we turn to the algorithm perspective. We propose the following general framework based on forward Euler discretization of the flows with finite-time convergence guarantee.

$$x_{k+1} = x_k + \eta F(x_k), \ \eta > 0 \tag{15}$$

where $F \in \{F_{q-\text{RGF}}, F_{q-\text{SGF}}\}$. We will show later, in Theorem 2, that the simple method leads, for small enough $\eta > 0$, to solutions that are $\mathcal{O}(\epsilon)$-close to the continuous-time flows. Also with a little abuse of notations, we will still use RGF/SGF to call the discrete-time algorithms induced by their corresponding continuous-time flows if the context is clear.

## 3.1 CLOSENESS OF THE DISCRETIZATION

We present here some convergence results of the proposed discretization. We first start with a proximity analysis, which aims at showing the closeness between the solutions of the continuous flows and the trajectories of their forward Euler discretization.

**Theorem 2** (Closeness of Discretization). *Suppose that $f : \mathbb{R}^n \to \mathbb{R}$ is continuously differentiable, locally $L_f$-Lipschitz, and $\mu$-gradient dominated of order $p \in (1, \infty)$ in a compact neighborhood $S$ of a strict local minimizer $x^\star \in \mathbb{R}^n$. Let $c > 0$ and $q \in (p, \infty]$. Then, for a given initial condition $x_0 \in S$, any maximal solution $x(t)$, $x(0) = x_0$, (in the sense of Filippov) to the $q$-RGF given by (2) or the $q$-SGF flow (3), there exists an arbitrarily small $\epsilon > 0$ such that the solution $x_k$ of any of the discrete algorithm (15), with sufficiently small $\eta > 0$, are $\epsilon$-close to $x(t)$, i.e., $\|x_k - x(t)\|_2 \leq \epsilon$ for $|t - k\eta| < \epsilon$, and s.t. the following bound holds*

$$\|f(x_k) - f(x^\star)\|_2 \leq L_f \epsilon + \left[ (f(x_0) - f(x^\star))^{(1-\alpha)} - \tilde{c}(1 - \alpha)\eta k \right]^{(1-\alpha)}, \ L_f > 0, \ k \leq k^\star,$$
$$\|f(x_k) - f(x^\star)\|_2 \leq L_f \epsilon, \ k > k^\star, \tag{16}$$

*where $\alpha = \frac{\theta}{\theta'}$, $\theta = \frac{p-1}{p}$, $\theta' = \frac{q-1}{q}$, $\tilde{c} = c \left( \left( \frac{p}{p-1} \right)^{\frac{p-1}{p}} \mu^{\frac{1}{p}} \right)^{\frac{1}{\theta'}}$ and $k^\star = \frac{(f(x_0) - f(x^\star))^{(1-\alpha)}}{\tilde{c}(1-\alpha)\eta}$.*

The analysis summarized in Theorem 2 is based on tools form hybrid control theory, and is detailed in Appendix D[2]. Theorem 2 shows that $\epsilon$-convergence of $x_k \to x^\star$ can be achieved in a finite number of steps upper bounded by $k^\star = \frac{(f(x_0) - f(x^\star))^{(1-\alpha)}}{\tilde{c}(1-\alpha)\eta}$. This is a preliminary convergence result aiming to show the existence of discrete solutions obtained via the proposed discretization algorithms, which are $\epsilon$-close to the continuous solutions of the finite-time flows. We also underline here that after $x_k$ reaches an $\epsilon$-neighborhood of $x^\star$, then $x_{k+1} \approx x_k$, $\forall k > k^\star$, since $x^\star$ is an equilibrium point of the continuous flows, i.e., $\Delta f(x^\star) = 0$, e.g. see Definition 2 in Appendix B.

## 3.2 CONVERGENCE RATES OF EULER DISCRETIZATION: GENERAL CASE

However, the bound (16) does not allow us to have a constructive practical value of the stepsize $\eta$, since the pair $(\epsilon, \eta)$ are implicitly related, for 'sufficiently small' $\eta$ (refer to the definition of $\epsilon$-closeness and the proof of Theorem 2 in Appendix D). To derive convergence rates of the proposed discretization with practical stepsize $\eta$, we need to introduce an extra assumption of $L$-Lipschitz smooth, as presented below.

**Assumption 2** (Lipschitz Smoothness of Order $q$). We assume the function $f$ is $L$-Lipschitz smooth of order $q \in (1, 2]$, i.e., for any $x, y \in \mathbb{R}^n$,

$$\|\nabla f(y) - \nabla f(x)\|_2 \leq L\|y - x\|_2^{q-1}. \tag{17}$$

**Remark 2.** The smoothness assumption is a very common setting in optimization algorithm literature (Nesterov, 2003; Beck, 2017). This assumption above is also called as $(L, q)$-Hölder continuity

---

[2]Note that there might be several ways of approaching this proof. For instance, one could follow the general results on stochastic approximation of set-valued dynamical systems, using the notion of perturbed solutions to differential inclusions presented in Benaïm et al. (2005).

(Devolder et al., 2014; Nesterov, 2015), it is easy to find that the condition will lead to the following property:

$$f(y) \le f(x) + \langle \nabla f(x), y - x \rangle + \frac{L}{q} \|y - x\|_2^q, \tag{18}$$

which is also called weak smoothness. When $q = 2$, the function will be Lipschitz smooth, so the above setting is a generalization of the common Lipschitz smoothness.

With these setting, now we will start our discussion from the general case, i.e., we have the access to the true gradient $\nabla f(x)$. The main result is presented in the following theorem, while we defer the proof to Appendix E. .

**Theorem 3** (Convergence Rate of the general Case). *Suppose that $f : \mathbb{R}^n \to \mathbb{R}$ is continuously differentiable, $\mu$-gradient dominated of order $q$ and $L$-Lipschitz smooth of order $q$ following Assumption 1 and 2, then running $q$-RGF (equation 2 and equation 15) for $K$ iterations with a stepsize $\eta = L^{\frac{1}{1-q}}$ satisfies that*

$$f(x_K) - f^\star \le \left(1 - \kappa^{\frac{1}{1-q}}\right)^K (f(x_0) - f^\star), \tag{19}$$

*and running $q$-SGF (equation 3 and equation 15) for $K$ iterations with $\eta = (nL)^{\frac{1}{1-q}}$ satisfies that*

$$f(x_K) - f^\star \le \left(1 - \left(n^{\frac{q}{2}}\kappa\right)^{\frac{1}{1-q}}\right)^K (f(x_0) - f^\star), \tag{20}$$

*where $n$ is the dimension number and $\kappa \triangleq \frac{L}{\mu}$. The corresponding iteration complexity are $\mathcal{O}\left(\kappa^{\frac{1}{q-1}} \ln \frac{1}{\epsilon}\right)$ and $\mathcal{O}\left(\left(n^{\frac{q}{2}}\kappa\right)^{\frac{1}{q-1}} \ln \frac{1}{\epsilon}\right)$, respectively.*

**Remark 3.** The above results show that with a constant stepsize, both the $q$-RGF and $q$-SGF can attain a linear convergence rate to reach the optimal point. Note that for the classical Lipschitz smooth case (i.e. $q = 2$), the RHS of the RGF result (equation 19) will reduce to $\left(1 - \kappa^{-1}\right)^K (f(x_0) - f^\star)$, which is the same as that in (Karimi et al., 2016, Theorem 1); while for SGF (equation 20), it will be $\left(1 - (n\kappa)^{-1}\right)^K (f(x_0) - f^\star)$, which recovers the result in (Beznosikov et al., 2020, Theorem 13)[3]. So we can conclude that our results extend the classical results in Lipschitz smoothness case.

### 3.3 Convergence Rates of Euler Discretization: Stochastic Case

Attaining the full gradient may be expensive in practical applications due to the large data size, while stochastic optimization algorithms are more applicable in such cases (Robbins & Monro, 1951; Bottou et al., 2018). So now we turn to discuss the complexity results when we have only the access to the unbiased estimator of the gradient. To formalize the discussion, we impose the following assumption on the gradient estimator, which is very common in stochastic optimization literature (Ghadimi & Lan, 2013; Bottou et al., 2018). .

**Assumption 3** (Unbiased Gradient Estimator). For any given $x \in \mathbb{R}^n$, we assume to have only the access of the unbiased gradient estimator, denoted as $g(x; \xi)$ where $\xi$ is a random variable, while we assume that it satisfies that $\mathbb{E}_\xi[g(x; \xi)] = \nabla f(x)$, $\mathbb{E}_\xi \|g(x; \xi) - \nabla f(x)\|_2^2 \le \sigma^2$.

**Stochastic $q$-RGF**    We present the convergence result of stochastic $q$-RGF in the following theorem, and defer the proof to Appendix F. For convenience, we denote $\psi \triangleq \frac{q}{q-1} \in [2, +\infty]$.

**Theorem 4** (Convergence Rate of the Stochastic RGF). *With the setting in Theorem 3, if we further assume for each $x \in \mathbb{R}^n$, we only have access to the unbiased estimator of the gradient in Assumption 3, then if we replace the gradient in $q$-RGF (equation 2) by its unbiased estimator $g(x) \triangleq \frac{1}{b(x)} \sum_{i=1}^{b(x)} g(x; \xi_i)$ with batch size $b(x) \in \mathbb{N}^+$. Then for $q$-RGF (equation 2 and equation 15) with $g(x)$, if we set $\eta = (\psi L)^{\frac{1}{1-q}}$, $b(x) \equiv \left(\frac{2 \cdot (2\sigma)^\psi \mu^{\frac{1}{1-q}}}{\epsilon}\right)^{\frac{2}{\psi}}$, then we have*

$$\mathbb{E}[f(x_K) - f^\star] \le \left(1 - \frac{2}{(2\psi)^\psi} \kappa^{\frac{1}{1-q}}\right)^K (f(x_0) - f^\star) + \frac{\epsilon}{2}. \tag{21}$$

---

[3]Note that $\frac{1}{n} \|\nabla f(x)\|_1^{-1} \text{sign}(\nabla f(x))$ is an $\frac{1}{n}$-approximate compressor of gradient $\nabla f(x)$ (Karimireddy et al., 2019), so we can apply (Beznosikov et al., 2020, Theorem 13) to recover the result.

**Stochastic $q$-SGF** The analysis of the Euler discretization of $q$-SGF in the stochastic case is a bit more complicated. Indeed, note that previously our discussion are based on $\ell_2$-norm. Here, however, following the argument in Balles et al. (2020), we change the gradient dominance and Lipschitz smoothness assumption to hold in $\ell_\infty$-norm (and its dual norm $\ell_1$).

**Assumption 4** (Gradient Dominance and Lipschitz Smoothness under $(\ell_1, \ell_\infty)$-norms)**.** We assume the function $f$ is $\mu$-gradient dominated of order $q$ and $L$-Lipschitz smooth of order $q$ under $\ell_\infty$-norm (and its dual norm $\ell_1$), i.e., for any $x, y \in \mathbb{R}^n$

$$\frac{q-1}{q}\|\nabla f(x)\|_1^{\frac{q}{q-1}} \geq \mu^{\frac{1}{q-1}}(f(x) - f^\star), \quad \|\nabla f(y) - \nabla f(x)\|_1 \leq L\|y - x\|_\infty^{q-1}. \quad (22)$$

Similarly the new smoothness condition above implies that $f(y) \leq f(x) + \langle \nabla f(x), y - x \rangle + \frac{L}{q}\|y - x\|_\infty^q$ ($\forall x, y \in \mathbb{R}^n$). And for the gradient estimator, we need the following assumption.

**Assumption 5** (SPB Unbiased Gradient Estimator (Safaryan & Richtárik, 2021))**.** For any given $x \in \mathbb{R}^n$, we assume to have only the access of the unbiased gradient estimator under $\ell_1$-norm, i.e., $\mathbb{E}_\xi[g(x;\xi)] = \nabla f(x)$, $\mathbb{E}_\xi\|g(x;\xi) - \nabla f(x)\|_1^2 \leq \sigma^2$; and for its mini-batch version $g(x)$ with batch size $b(x)$, we further assume that it satisfies *success probability bounds (SPB) property*, i.e., for each component of $g(x)$ and $\nabla f(x)$ (denoted as $g_i(x)$ and $\nabla_i f(x)$), we have $\exists\, p^* \in (\frac{1}{2}, 1]$ such that

$$\mathbb{P}(\text{sign}(\nabla_i f(x)) = \text{sign}(g_i(x))) \geq p^* > \frac{1}{2}, \quad \forall\, i \in \{1, 2, \cdots, n\}. \quad (23)$$

**Remark 4.** Note that similar assumptions under $\ell_\infty$-norm (and its dual norm $\ell_1$) are widely adapted in the literature of sign-based algorithms, e.g., Karimi et al. (2016); Bernstein et al. (2018a;b); Balles et al. (2020). Here the extra SPB assumption, proposed in Safaryan & Richtárik (2021), is very important in the convergence analysis of stochastic sign-based algorithms. An intuitive understanding to justify SPB is that, with a large batch size, the output estimator $g(x)$ will be very similar to $\nabla f(x)$, so the corresponding probability above should be higher. For more details on the rationality of the assumption, we refer the reader to Safaryan & Richtárik (2021) for further discussions.

With the above preparation, we summarize the result of stochastic SGF in the following theorem, while deferring the proof to Appendix G.

**Theorem 5** (Convergence Rate of the Stochastic SGF)**.** *Suppose the function $f$ satisfies the setting in Assumption 4 above, then for $q$-SGF (equation 3 and equation 15) with an unbiased gradient estimator $g(x)$ of batch size $b(x) \in \mathbb{N}^+$, if we assume the gradient estimator $g(x)$ satisfies Assumption 5, and set $\eta = \left(\frac{L2^{\psi-1}}{2p^*-1}\right)^{\frac{1}{1-q}}$ and $b(x_k) \equiv \left(\frac{2\mu^{\frac{1}{1-q}}}{q\epsilon}\right)^{\frac{2}{\psi}}(2p^*-1)^{-4}\sigma^2$, it satisfies that*

$$\mathbb{E}[f(x_K) - f^\star] \leq \left(1 - (2p^*-1)^\psi(\kappa2^{\psi-1})^{\frac{1}{1-q}}\right)^K (f(x_0) - f^\star) + \frac{\epsilon}{2}. \quad (24)$$

**Remark 5.** The results in Theorem 4 and Theorem 5 shows that both stochastic RGF and SGF will drive the function value to linearly converge to the $\mathcal{O}(\epsilon)$-neighborhood of the optimal value. As a comparison on stochastic RGF, Lei et al. (2019) studied the convergence guarantee of stochastic gradient descent (SGD), which can be viewed as stochastic 2-RGF, under a similar setting. Also for stochastic SGF, we notice that recently Li et al. (2021) provided a similar convergence result of stochastic SGF in the classical setting (i.e., $q = 2$), their result also requires an lower bound on the probability of estimation correctness, which corresponds to the SPB assumption above. Our result can be viewed as an extension of their work.

## 4 Numerical Experiments

In this section, we will use several experiments to verify the effectiveness of our proposed algorithms, more details will be presented in Appendix H.

### 4.1 Numerical Experiments on Academic Examples

Let us show first on a simple numerical example that the acceleration in convergence, proven in continuous time for certain range of the hyparmeters, can translate to some convergence acceleration in discrete time.

**Example 1**   Consider the following function: $f : \mathbb{R} \to \mathbb{R}$ and $f(x) \triangleq \frac{1}{q}|x|^q$, which can be shown to be $(q-1)^{q-1}$-gradient dominated of order $q$, and 2-Lipschitz smooth of order $q$ when $1 < q \leq 2$ (Lei & Ying, 2020; Romero & Benosman, 2020). Here we set $q = 1.95$ for the objective function. The test results of the proposed RGF and SGF algorithms compared with other algorithms is presented in Figure 1 below. The figure shows both sensitivities to stepsizes and performances of algorithms, measured by the optimality gap $f(x) - f^*$ where $f^*$ is the optimal value which is 0. We can find that $q$-RGF and $q$-SGF, comparing to the classical one (2-RGF) and an over-large one (10-RGF), can accommodate larger stepsize and attain better performances with the best-tuned stepsize[4]. Also we further plot the theoretical bounds provided in Theorem 3 (in red), we can find that the proposed algorithms attain linear convergence patterns and their optimal stepsize choice is close to those in Theorem 3, so the experiment results matches the theory well in the example.

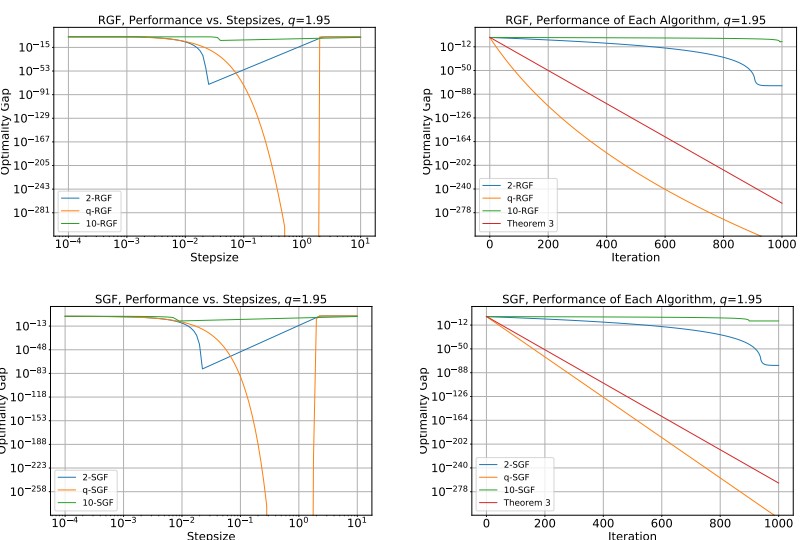

Figure 1: Results of discretization of RGF and SGF with different $q$ on Example 1

**Example 2**   We consider the Rosenbrock function $f : \mathbb{R}^2 \to \mathbb{R}$, given by $f(x_1, x_2) = (a - x_1)^2 + b(x_2 - x_1^2)^2$, with parameters $a, b \in \mathbb{R}$, which is locally Lipschitz smooth. This function admits exactly one stationary point $(x_1^\star, x_2^\star) = (a, a^2)$ for $b \geq 0$, and is locally strongly convex, hence locally satisfies gradient dominance of order $p = 2$, which allows us to select $q > 2$ in $q$-RGF and $q$-SGF to achieve finite-time convergence in conitnuous-time. We report in Figure 2 the mean performance of forward-Euler discretization for $q$-RGF and $q$-SGF, with fixed step size[5], for several values of $q$, for 10 random initial conditions in $[0, 2]$. We observe that, as expected from the continuous flow analysis, for $q$ close to 2, $q$-RGF behaves similar to GD in terms of convergence rate, whereas for $q > 2$ the finite-time convergence in continuous time translates to some acceleration in the associated discretization algorithm as well. Similarly for $q$-SGF, $q$ closer to 2 translates to less accelerated algorithms, with a behavior similar to GD, whereas larger $q$ values lead to accelerated convergence.

### 4.2   NUMERICAL EXPERIMENTS ON REAL-WORLD DATA

We report here the results of our experiments using deep neural network (DNN) training on the SVHN dataset. Note that, we use PyTorch platform to conduct all the tests reported in this paper. Both CPU and GPU tests have been performed. The results reported in this section are CPU tests, whereas GPU tests are reported in Appendix H. We underline here that the DNNs are non-convex globally, however, one could assume at least local convexity, hence local gradient dominance of order $p = 2$, thus, we will select $q > 2$ in our experiments (see (Remark 7, Appendix H) for more explanations on the choice of $q$).

---

[4]The experiment results show that $q$-RGF and $q$-SGF will drive the function value to exactly 0 with certain stepsizes, which generates plots that may exceed the lower limit in figures.

[5]We did multiple iterations to find the best step size for each algorithm (best values where between $10^{-4}$ and $10^{-2}$ depending on the algorithm). Details of the step size for each test are given in Appendix H.

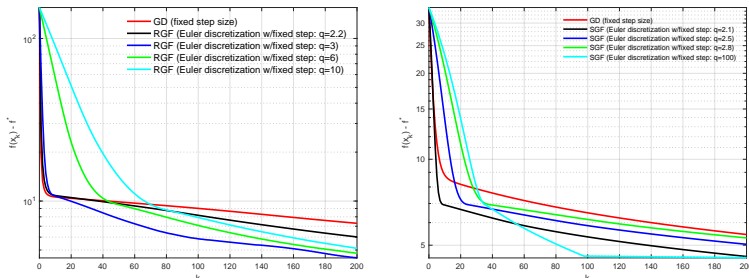

Figure 2: Results of Discretization of RGF and SGF with Different $q$ on Example 2

### 4.2.1 EXPERIMENTS ON SVHN DATASET

We test the proposed algorithms to train the same VGG16 CNN model with cross entropy loss on the SVHN dataset. We divided the dataset into a training set of 74 batches with 1000 images each, and a test set of 27 batches of 1000 images each, and ran 20 epochs of training over all the training batches. We tested the discretization of $q$-RGF ($c = 1$, $q = 2.1$, $\eta = 0.04$), and of $q$-SGF ($c = 10^{-3}$, $q = 2.1$, $\eta = 0.04$) against classical gradient descent (GD), and Adam[6]. Note from Figures 3, 4 it is clear that $q$-RGF and $q$-SGF give a good performance in terms of convergence speed, and final test performance $93\%$. We can also observe in Figure 4 that $q$-SGF, and $q$-RGF converge faster ($40\ min$ lead in average) than SGD and Adam for these tests, and reach an overall similar performance on the test-set. Additional numerical results, including GPU runs, can be found in Appendix H.

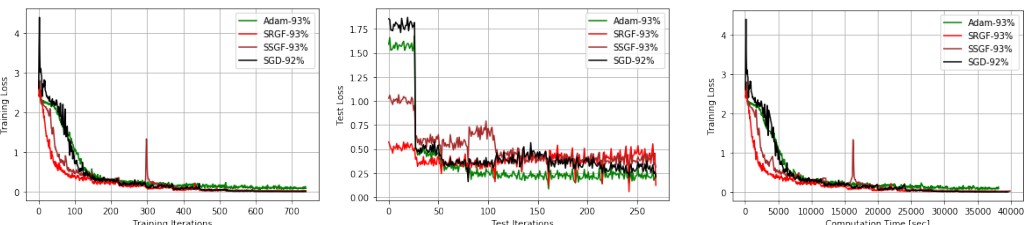

Figure 3: Losses for several optimization algorithms- SVHN: Train loss (left), test loss (right)

Figure 4: Training loss vs. computation time for several optimization algorithms- VGG-16-SVHN

## 5 CONCLUSION

We presetned some connections between optimization algorithms and continuous-time representations (dynamical systems) via discretization. We then reviewed two families of non-Lipschitz or discontinuous first-order optimization flows for continuous-time optimization, namely the $q$-RGF and $q$-SGF, whose distinguishing characteristic is their finite-time convergence. We then proposed a forward Euler discretization of these flows. Based on tools from hybrid systems control theory, we proved a closeness convergence bound for these algorithms, and then proposed several convergence rates in the deterministic and the stochastic setting. Finally, we conducted numerical experiments on a known deep neural network benchmark, which showed that the proposed discrete algorithms can outperform some state of the art algorithms, when tested on large DNN models.

---

[6]We also tested Adaptive gradient (AdaGrad), per-dimension learning rate method for gradient descent (AdaDelta), and Root Mean Square Propagation (RMSprop). However, since their performance was not competitive we decided not to report the plots to avoid overloading the figures.

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

## A    Discontinuous Systems and Differential Inclusions

Recall that for an initial value problem (IVP)

$$\dot{x}(t) = F(x(t)) \tag{25a}$$
$$x(0) = x_0 \tag{25b}$$

with $F : \mathbb{R}^n \to \mathbb{R}^n$, the typical way to check for existence of solutions is by establishing continuity of $F$. Likewise, to establish uniqueness of the solution, we typically seek Lipschitz continuity. When $F$ is discontinuous, we may understand (25a) as the Filippov differential inclusion

$$\dot{x}(t) \in \mathcal{K}[F](x(t)), \tag{26}$$

where $\mathcal{K}[F] : \mathbb{R}^n \rightrightarrows \mathbb{R}^n$ denotes the Filippov set-valued map given by

$$\mathcal{K}[F](x) \triangleq \bigcap_{\delta > 0} \bigcap_{\mu(S)=0} \overline{\mathrm{co}}\, F(B_\delta(x) \setminus S), \tag{27}$$

where $\mu$ denotes the usual Lebesgue measure and $\overline{\mathrm{co}}$ the convex closure, i.e. closure of the convex hull $\overline{\mathrm{co}}$. For more details, see Paden & Sastry (1987). We can generalize (26) to the differential inclusion Bacciotti & Ceragioli (1999)

$$\dot{x}(t) \in \mathcal{F}(x(t)), \tag{28}$$

where $\mathcal{F} : \mathbb{R}^n \rightrightarrows \mathbb{R}^n$ is some set-valued map.

**Definition 1** (Carathéodory/Filippov solutions). We say that $x : [0, \tau) \to \mathbb{R}^n$ with $0 < \tau \leq \infty$ is a *Carathéodory solution* to (28) if $x(\cdot)$ is absolutely continuous and (28) is satisfied a.e. in every compact subset of $[0, \tau)$. Furthermore, we say that $x(\cdot)$ is a *maximal* Carathéodory solution if no other Carathéodory solution $x'(\cdot)$ exists with $x = x'|_{[0,\tau)}$. If $\mathcal{F} = \mathcal{K}[F]$, then Carathéodory solutions are referred to as *Filippov solutions*.

For a comprehensive overview of discontinuous systems, including sufficient conditions for existence (Proposition 3) and uniqueness (Propositions 4 and 5) of Filippov solutions, see the work of Cortés (2008). In particular, it can be established that Filippov solutions to (25) exist, provided that the following assumption (Assumption 6) holds.

**Assumption 6** (Existence of Filippov solutions). $F : \mathbb{R}^n \to \mathbb{R}^n$ is defined almost everywhere (a.e.) and is Lebesgue-measurable in a non-empty open neighborhood $U \subset \mathbb{R}^n$ of $x_0 \in \mathbb{R}^n$. Further, $F$ is locally essentially bounded in $U$, *i.e.*, for every point $x \in U$, $F$ is bounded a.e. in some bounded neighborhood of $x$.

More generally, Carathéodory solutions to (28) exist (now with arbitrary $x_0 \in \mathbb{R}^n$), provided that the following assumption (Assumption 7) holds.

**Assumption 7** (Existence of Carathéodory solutions). $\mathcal{F} : \mathbb{R}^n \rightrightarrows \mathbb{R}^n$ has nonempty, compact, and convex values, and is *upper semi-continuous*.

Filippov & Arscott (1988) proved that, for the Filippov set-valued map $\mathcal{F} = \mathcal{K}[F]$, Assumptions 6 and 7 are equivalent (with arbitrary $x_0 \in \mathbb{R}^n$ in Assumption 6).

Uniqueness of the solution requires further assumptions. Nevertheless, we can characterize the Filippov set-valued map in a similar manner to Clarke's generalized gradient, as seen in the following proposition.

**Proposition 1** (Theorem 1 of Paden & Sastry (1987)). *Under Assumption 6, we have*

$$\mathcal{K}[F](x) = \left\{ \lim_{k \to \infty} F(x_k) : x_k \in \mathbb{R}^n \setminus (\mathcal{N}_F \cup S) \text{ s.t. } x_k \to x \right\} \tag{29}$$

*for some (Lebesgue) zero-measure set $\mathcal{N}_F \subset \mathbb{R}^n$ and any other zero-measure set $S \subset \mathbb{R}^n$. In particular, if $F$ is continuous at a fixed $x$, then $\mathcal{K}[F](x) = \{F(x)\}$.*

For instance, for the GF (1), we have $\mathcal{K}[-\nabla f](x) = \{-\nabla f(x)\}$ for every $x \in \mathbb{R}^n$, provided that $f$ is continuously differentiable. Furthermore, if $f$ is only locally Lipschitz continuous and regular (see Definition 3 of Appendix B), then $\mathcal{K}[-\nabla f](x) = -\partial f(x)$, where

$$\partial f(x) \triangleq \left\{ \lim_{k \to \infty} \nabla f(x_k) : x_k \in \mathbb{R}^n \setminus \mathcal{N}_f \text{ s.t. } x_k \to x \right\} \tag{30}$$

denotes Clarke's generalized gradient Clarke (1981) of $f$, with $\mathcal{N}_f$ denoting the zero-measure set over which $f$ is not differentiable (Rademacher's theorem). It can be established that $\partial f$ coincides with the subgradient of $f$, provided that $f$ is convex. Therefore, the GF (1) interpreted as Filippov differential inclusion may also be seen as a continuous-time variant of subgradient descent methods.

## B  FINITE-TIME STABILITY OF DIFFERENTIAL INCLUSIONS

We are now ready to focus on extending some notions from traditional Lipschitz continuous systems to differential inclusions.

**Definition 2.** We say that $x^\star \in \mathbb{R}^n$ is an *equilibrium* of (28) if $x(t) \equiv x^\star$ on some small enough non-degenerate interval is a Carathéodory solution to (28). In other words, if and only if $0 \in \mathcal{F}(x^\star)$. We say that (28) is *(Lyapunov) stable* at $x^\star \in \mathbb{R}^n$ if, for every $\varepsilon > 0$, there exists some $\delta > 0$ such that, for every maximal Carathéodory solution $x(\cdot)$ of (28), we have $\|x_0 - x^\star\|_2 < \delta \implies \|x(t) - x^\star\|_2 < \varepsilon$ for every $t \geq 0$ in the interval where $x(\cdot)$ is defined. Note that, under Assumption 7, if (28) is stable at $x^\star$, then $x^\star$ is an equilibrium of (28) Bacciotti & Ceragioli (1999). Furthermore, we say that (28) is *(locally and strongly) asymptotically stable* at $x^\star \in \mathbb{R}^n$ if is stable at $x^\star$ and there exists some $\delta > 0$ such that, for every maximal Carathéodory solution $x : [0, \tau) \to \mathbb{R}^n$ of (28), if $\|x_0 - x^\star\|_2 < \delta$ then $x(t) \to x^\star$ as $t \to \tau$. Finally, (28) is *(locally and strongly) finite-time stable* at $x^\star$ if it is asymptotically stable and there exists some $\delta > 0$ and $T : B_\delta(x^\star) \to [0, \infty)$ such that, for every maximal Carathéodory solution $x(\cdot)$ of (28) with $x_0 \in B_\delta(x^\star)$, we have $\lim_{t \to T(x_0)} x(t) = x^\star$.

We will now construct a Lyapunov-based criterion adapted from the literature of finite-time stability of Lipschitz continuous systems.

**Lemma 1.** *Let $\mathcal{E}(\cdot)$ be an absolutely continuous function satisfying the differential inequality*

$$\dot{\mathcal{E}}(t) \leq -c\, \mathcal{E}(t)^\alpha \tag{31}$$

*a.e. in $t \geq 0$, with $c, \mathcal{E}(0) > 0$ and $\alpha < 1$. Then, there exists some $t^\star > 0$ such that $\mathcal{E}(t) > 0$ for $t \in [0, t^\star)$ and $\mathcal{E}(t^\star) = 0$. Furthermore, $t^\star > 0$ can be bounded by*

$$t^\star \leq \frac{\mathcal{E}(0)^{1-\alpha}}{c(1-\alpha)}, \tag{32}$$

*with this bound tight whenever (31) holds with equality. In that case, but now with $\alpha \geq 1$, then $\mathcal{E}(t) > 0$ for every $t \geq 0$, with $\lim_{t \to \infty} \mathcal{E}(t) = 0$. This will be represented by $t^\star = \infty$, with $\mathcal{E}(\infty) \triangleq \lim_{t \to \infty} \mathcal{E}(t)$.*

*Proof.* Suppose that $\mathcal{E}(t) > 0$ for every $t \in [0, T]$ with $T > 0$. Let $t^\star$ be the supremum of all such $T$'s, thus satisfying $\mathcal{E}(t) > 0$ for every $t \in [0, t^\star)$. We will now investigate $\mathcal{E}(t^\star)$. First, by continuity of $\mathcal{E}$, it follows that $\mathcal{E}(t^\star) \geq 0$. Now, by rewriting

$$\dot{\mathcal{E}}(t) \leq -c\, \mathcal{E}(t)^\alpha \iff \frac{\mathrm{d}}{\mathrm{d}t}\left[\frac{\mathcal{E}(t)^{1-\alpha}}{1-\alpha}\right] \leq -c, \tag{33}$$

a.e. in $t \in [0, t^\star)$, we can thus integrate to obtain

$$\frac{\mathcal{E}(t)^{1-\alpha}}{1-\alpha} - \frac{\mathcal{E}(0)^{1-\alpha}}{1-\alpha} \leq -ct, \tag{34}$$

everywhere in $t \in [0, t^\star)$, which in turn turn leads to

$$\mathcal{E}(t) \leq [\mathcal{E}(0)^{1-\alpha} - c(1-\alpha)t]^{1/(1-\alpha)} \tag{35}$$

and

$$t \leq \frac{\mathcal{E}(0)^{1-\alpha} - \mathcal{E}(t)^{1-\alpha}}{c(1-\alpha)} \leq \frac{\mathcal{E}(0)^{1-\alpha}}{c(1-\alpha)}, \tag{36}$$

where the last inequality follows from $\mathcal{E}(t) > 0$ for every $t \in [0, t^\star)$. Taking the supremum in (36) then leads to the upper bound (32). Finally, we conclude that $\mathcal{E}(t^\star) = 0$, since $\mathcal{E}(t^\star) > 0$ is impossible given that it would mean, due to continuity of $\mathcal{E}$, that there exists some $T > t^\star$ such that $\mathcal{E}(t) > 0$ for every $t \in [0, T]$, thus contradicting the construction of $t^\star$.

Finally, notice that if $\mathcal{E}$ is such that (31) holds with equality, then (35) and the first inequality in (36) hold with equality as well. The tightness of the bound (32) thus follows immediately. Furthermore, notice that if $\alpha \geq 1$, and $\mathcal{E}$ is a tight solution to the differential inequality (31), *i.e.* $\mathcal{E}(t) = [\mathcal{E}(0)^{1-\alpha} - c(1-\alpha)t]^{1/(1-\alpha)}$, then clearly $\mathcal{E}(t) > 0$ for every $t \geq 0$ and $\mathcal{E}(t) \to 0$ as $t \to \infty$. ∎

Cortés & Bullo (2005) proposed (Proposition 2.8) a Lyapunov-based criterion to establish finite-time stability of discontinuous systems, which fundamentally coincides with our Lemma 1 for the particular choice of exponent $\alpha = 0$. Their proposition was, however, directly based on Theorem 2 of Paden & Sastry (1987). Later, Cortés (2006) proposed a second-order Lyapunov criterion, which, on the other hand, fundamentally translates to $\mathcal{E}(t) \triangleq V(x(t))$ being strongly convex. Finally, Hui et al. (2009) generalized Proposition 2.8 of Cortés & Bullo (2005) in their Corollary 3.1, to establish semistability. Indeed, these two results coincide for isolated equilibria.

We now present a novel result that generalizes the aforementioned first-order Lyapunov-based results, by exploiting our Lemma 1. More precisely, given a Laypunov candidate function $V(\cdot)$, the objective is to set $\mathcal{E}(t) \triangleq V(x(t))$, and we aim to check that the conditions of Lemma 1 hold. To do this, and assuming $V$ to be locally Lipschitz continuous, we first borrow and adapt from Bacciotti & Ceragioli (1999) the definition of *set-valued time derivative* of $V : \mathcal{D} \to \mathbb{R}$ w.r.t. the differential inclusion (28), given by

$$\dot{V}(x) \triangleq \{a \in \mathbb{R} : \exists v \in \mathcal{F}(x) \text{ s.t. } a = p \cdot v, \forall p \in \partial V(x)\}, \tag{37}$$

for each $x \in \mathcal{D}$. Notice that, under Assumption 7 for Filippov differential inclusions $\mathcal{F} = \mathcal{K}[F]$, the set-valued time derivative of $V$ thus coincides with with the set-valued Lie derivative $\mathcal{L}_F V(\cdot)$. Indeed, more generally $\dot{V}$ could be seen as a set-valued Lie derivative $\mathcal{L}_{\mathcal{F}} V$ w.r.t. the set-valued map $\mathcal{F}$.

**Definition 3.** $V(\cdot)$ is said to be *regular* if every directional derivative, given by

$$V'(x; v) \triangleq \lim_{h \to 0} \frac{V(x + h\,v) - V(x)}{h}, \tag{38}$$

exists and is equal to

$$V^{\circ}(x; v) \triangleq \limsup_{x' \to x\ h \to 0^+} \frac{V(x' + h\,v) - V(x')}{h}, \tag{39}$$

known as *Clarke's upper generalized derivative* Clarke (1981).

In practice, regularity is a fairly mild and easy to guarantee condition. For instance, it would suffice that $V$ is convex or continuously differentiable to ensure that it is Lipschitz and regular.

**Assumption 8.** $V : \mathcal{D} \to \mathbb{R}$ is locally Lipscthiz continuous and regular, with $\mathcal{D} \subseteq \mathbb{R}^n$ open.

Under Assumption 8, Clarke's generalized gradient

$$\partial V(x) \triangleq \{p \in \mathbb{R}^n : V^{\circ}(x; v) \geq p \cdot v, \forall v \in \mathbb{R}^n\} \tag{40}$$

is non-empty for every $x \in \mathcal{D}$, and is also given by

$$\partial V(x) = \left\{ \lim_{k \to \infty} \nabla V(x_k) : x_k \in \mathbb{R}^n \setminus \mathcal{N}_V \text{ s.t. } x_k \to x \right\}, \tag{41}$$

where $\mathcal{N}_V$ denotes the set of points in $\mathcal{D} \subseteq \mathbb{R}^n$ where $V$ is not differentiable (Rademacher's theorem) Clarke (1981).

Through the following lemma (Lemma 2), we can formally establish the correspondence between the set-valued time-derivative of $V$ and the derivative of the energy function $\mathcal{E}(t) \triangleq V(x(t))$ associated with an arbitrary Carathéodory solution $x(\cdot)$ to the differential inclusion (28).

**Lemma 2** (Lemma 1 of Bacciotti & Ceragioli (1999)). *Under Assumption 8, given any Carathéodory solution* $x : [0, \tau) \to \mathbb{R}^n$ *to (28), then* $\mathcal{E}(t) \triangleq V(x(t))$ *is absolutely continuous and* $\dot{\mathcal{E}}(t) = \frac{\mathrm{d}}{\mathrm{d}t} V(x(t)) \in \dot{V}(x(t))$ *a.e. in* $t \in [0, \tau)$.

We are now ready to state and prove our Lyapunov-based sufficient condition for finite-time stability of differential inclusions.

**Theorem 6.** *Suppose that Assumptions 7 and 8 hold for some set-valued map $\mathcal{F} : \mathbb{R}^n \rightrightarrows \mathbb{R}^n$ and some function $V : \mathcal{D} \to \mathbb{R}$, where $\mathcal{D} \subseteq \mathbb{R}^n$ is an open and positively invariant neighborhood of a point $x^\star \in \mathbb{R}^n$. Suppose that $V$ is positive definite w.r.t. $x^\star$ and that there exist constants $c > 0$ and $\alpha < 1$ such that*

$$\sup \dot{V}(x) \leq -c\, V(x)^\alpha \tag{42}$$

*a.e. in $x \in \mathcal{D}$. Then, (28) is finite-time stable at $x^\star$, with settling time upper bounded by*

$$t^\star \leq \frac{V(x_0)^{1-\alpha}}{c(1-\alpha)}, \tag{43}$$

*where $x(0) = x_0$. In particular, any Carathéodory solution $x(\cdot)$ with $x(0) = x_0 \in \mathcal{D}$ will converge in finite time to $x^\star$ under the upper bound (43). Furthermore, if $\mathcal{D} = \mathbb{R}^n$, then (28) is globally finite-time stable. Finally, if $\dot{V}(x)$ is a singleton a.e. in $x \in \mathcal{D}$ and (42) holds with equality, then the bound (43) is tight.*

*Proof.* Note that, by Proposition 1 of Bacciotti & Ceragioli (1999), we know that (28) is Lyapunov stable at $x^\star$. All that remains to be shown is local convergence towards $x^\star$ (which must be an equilibrium) in finite time. Indeed, given any maximal solution $x : [0, t^\star) \to \mathbb{R}^n$ to (28) with $x(0) = x_0 \neq x^\star$, we know by Lemma 2, that $\mathcal{E}(t) = V(x(t))$ is absolutely continuous with $\dot{\mathcal{E}}(t) \in \dot{V}(x(t))$ a.e. in $t \in [0, t^\star]$. Therefore, we have

$$\dot{\mathcal{E}}(t) \leq \sup \dot{V}(x(t)) \leq -c\, V(x(t))^\alpha = -c\, \mathcal{E}(t)^\alpha \tag{44}$$

a.e. in $t \in [0, t^\star]$. Since $\mathcal{E}(0) = V(x_0) > 0$, given that $x_0 \neq x^\star$, the result then follows by invoking Lemma 1 and noting that $\mathcal{E}(t^\star) = 0 \iff V(t^\star, x(t^\star)) = 0 \iff x(t^\star) = x^\star$. ∎

Finite-time stability still follows without Assumption 7, provided that $x^\star$ is an equilibrium of (28). In practical terms, this means that trajectories starting arbitrarily close to $x^\star$ may not actually exist, but nevertheless there exists a neighborhood $\mathcal{D}$ of $x^\star$ over which, any trajectory $x(\cdot)$ that indeed exists and starts at $x(0) = x_0 \in \mathcal{D}$ must converge in finite time to $x^\star$, with settling time upper bounded by $T(x_0)$ (the bound still tight in the case that (42) holds with equality).

## C  PROOF OF THEOREM 1

Let us focus on the $q$-RGF (2) (the case of $q$-SGF (3) follows exactly the same steps) with the candidate Lyapunov function $V \triangleq f - f(x^\star)$. Clearly, $V$ is Lipschitz continuous and regular (given that it is continuously differentiable). Furthermore, $V$ is positive definite w.r.t. $x^\star$.

Notice that, due to the dominated gradient assumption, $x^\star$ must be an isolated stationary point of $f$. To see this, notice that, if $x^\star$ were not an isolated stationary point, then there would have to exist some $\tilde{x}^\star$ sufficiently near $x^\star$ such that $\tilde{x}^\star$ is both a stationary point of $f$, and satisfies $f(\tilde{x}^\star) > f(x^\star)$, since $x^\star$ is a strict local minimizer of $f$. But then, we would have

$$0 = \frac{p-1}{\psi}\|\nabla f(\tilde{x}^\star)\|_2^{\frac{\psi}{p-1}} \geq \mu^{\frac{1}{p-1}}(f(\tilde{x}^\star) - f(x^\star)) > 0, \tag{45}$$

and subsequently $0 > 0$, which is absurd.

Therefore, $F(x) \triangleq -c\nabla f(x)/\|\nabla f(x)\|_2^{\frac{q-2}{q-1}}$ is continuous for every $x \in \mathcal{D} \setminus \{0\}$, for some small enough open neighborhood $\mathcal{D}$ of $x^\star$. Let us assume that $\mathcal{D}$ is positively invariant w.r.t. (2), which can be achieved, for instance, by replacing $\mathcal{D}$ with its intersection with some small enough strict sublevel set of $f$. Notice that $\|F(x)\|_2 = c\|\nabla f(x)\|_2^{\frac{1}{q-1}}$ with $q \in (p, \infty] \subset (1, \infty]$, *i.e.*, $\frac{1}{q-1} \in [0, \infty)$. If $q = \infty$, which results in the normalized gradient flow $\dot{x} = -\frac{\nabla f(x)}{\|\nabla f(x)\|_2}$ proposed by Cortés (2006), then $\|F(x)\|_2 = c > 0$. We can thus show that $F(x)$ is discontinuous at $x = 0$ for $q = \infty$. On the other hand, if $q \in (p, \infty) \subset (1, \infty)$, then we have $\|F(x)\|_2 \to 0$ as $x \to x^\star$, and thus $F(x)$ is continuous (but not Lipschitz) at $x = x^\star$. Regardless, we may freely focus exclusively on $\mathcal{D} \setminus \{x^\star\}$ since $\{x^\star\}$ is obviously a zero-measure set.

Let $\mathcal{F} \triangleq \mathcal{K}[F]$. We thus have, for each $x \in \mathcal{D} \setminus \{x^\star\}$,

$$\sup \dot{V}(x) = \sup\{a \in \mathbb{R} : \exists v \in \mathcal{F}(x) \text{ s.t. } a = p \cdot v, \forall p \in \partial V(x)\} \tag{46a}$$

$$= \sup\{\nabla V(x) \cdot v : v \in \mathcal{F}(x)\} \tag{46b}$$

$$= \nabla V(x) \cdot F(x) \tag{46c}$$

$$= -c\|\nabla f(x)\|_2^{2 - \frac{q-2}{q-1}} \tag{46d}$$

$$= -c\|\nabla f(x)\|_2^{\frac{1}{\theta'}} \tag{46e}$$

$$\leq -c[C(f(x) - f(x^\star))^\theta]^{\frac{1}{\theta'}} \tag{46f}$$

$$= -cC^{\frac{1}{\theta'}} V(x)^{\frac{\theta}{\theta'}}. \tag{46g}$$

Since $\frac{\theta}{\theta'} < 1$, given that $s > 1 \mapsto \frac{s-1}{s}$ is strictly increasing, then the conditions of Theorem 6 are satisfied. In particular, we have finite-time stability at $x^\star$ with a settling time $t^\star$ upper bounded by

$$t^\star \leq \frac{(f(x_0) - f(x^\star))^{1 - \frac{\theta}{\theta'}}}{cC^{\frac{1}{\theta'}}\left(1 - \frac{\theta}{\theta'}\right)} \leq \frac{(\|\nabla f(x_0)\|_2/C)^{\frac{1}{\theta}\left(1 - \frac{\theta}{\theta'}\right)}}{cC^{\frac{1}{\theta'}}\left(1 - \frac{\theta}{\theta'}\right)} = \frac{\|\nabla f(x_0)\|_2^{\frac{1}{\theta} - \frac{1}{\theta'}}}{cC^{\frac{1}{\theta}}\left(1 - \frac{\theta}{\theta'}\right)} \tag{47}$$

for each $x_0 \in \mathcal{D}$, which completes the proof.

## D  PROOF OF THEOREM 2

To prove Theorem 2, we borrow some tools and results from hybrid control systems theory. Hybrid control systems are characterized by continuous flows with discrete jumps between the continuous flows. They are often modeled by differential inclusions added to discrete mappings to model the jumps between the differential inclusions. We see the case of the optimization flows proposed here as a simple case of a hybrid systems with one differential inclusion, with a possible jump or discontinuity at the optimum. Based on this, we will use the tools and results of Sanfelice & Teel (2010), which study how a certain class of hybrid systems behave after discretization with a certain class of discretization algorithms. In other words, Sanfelice & Teel (2010) quantifies, under some conditions, how close are the solutions of the discretized hybrid dynamical system to the solutions of the original hybrid system.

In this section we will denote the differential inclusion of the continuous optimization flow by $\mathcal{F} : \mathbb{R}^n \rightrightarrows \mathbb{R}^n$, and its discretization in time by $\mathcal{F}_{\mathrm{d}} : \mathbb{R}^n \rightrightarrows \mathbb{R}^n$. We first recall a definition, which we will adapt from the general case of jumps between multiple differential inclusions (Definition 3.2, Sanfelice & Teel (2010)) to our case of one differential inclusion or flow.

**Definition 4.** (($T, \epsilon$)-closeness). Given $T > 0$, $\epsilon > 0$, $\eta > 0$, two solutions $x_t : [0, T] \to \mathbb{R}^n$, and $x_k : \{0, 1, 2, ...\} \to \mathbb{R}^n$ are ($T, \epsilon$)-close if:
(a) for all $t \leq T$ there exists $k \in \{1, 2, ...\}$ such that $|t - k\eta| < \epsilon$, and $\|x_t(t) - x_k(k)\|_2 < \epsilon$,
(b) for all $k \in \{1, 2, ...\}$ there exists $t \leq T$ such that $|t - k\eta| < \epsilon$, and $\|x_t(t) - x_k(k)\|_2 < \epsilon$.

Next, we will recall Theorem 5.2 in Sanfelice & Teel (2010), while adapting it to our special case of a hybrid system with one differential inclusion[7].

**Theorem 7.** *(Closeness of continuous and discrete solutions on compact domains) Consider the differential inclusion*

$$\dot{X}(t) \in \mathcal{F}(X(t)), \tag{48}$$

*for a given set-valued mapping $\mathcal{F} : \mathbb{R}^m \rightrightarrows \mathbb{R}^m$ assumed to be outer semicontinuous, locally bounded, nonempty, and with convex values for every $x \in \mathcal{C}$, for some closed set $\mathcal{C} \subseteq \mathbb{R}^m$. Consider the discrete-time system represented by the flow $\mathcal{F}_{\mathrm{d}} : \mathbb{R}^n \rightrightarrows \mathbb{R}^n$, such that, for each compact set $K \subset \mathbb{R}^n$, there exists $\rho \in \mathcal{K}_\infty$, and $\eta^\star > 0$ such that for each $x \in K$ and each $\eta \in (0, \eta^\star]$,*

$$\mathcal{F}_{\mathrm{d}}(x) \subset x + \eta \operatorname{con}\mathcal{F}(x + \rho(\eta)\mathbb{B}) + \eta\rho(\eta)\mathbb{B}. \tag{49}$$

---

[7]A set-valued mapping $\mathcal{F} : \mathbb{R}^n \rightrightarrows \mathbb{R}^n$ is *outer semicontinuous* if for each sequence $\{x_i\}_{i=1}^\infty$ converging to a point $x \in \mathbb{R}^n$ and each sequence $y_i \in \mathcal{F}(x_i)$ converging to a point $y$, it holds that $y \in \mathcal{F}(x)$. It is *locally bounded* if, for each point $x \in \mathbb{R}^n$, there exists compact sets $K, K' \subset \mathbb{R}^n$ such that $x \in K$ and $\mathcal{F}(K) \triangleq \cup_{x \in K}\mathcal{F}(x) \subset K'$. In what follows, we use the following notations: Given a set $A$, $\operatorname{con}A$ denotes the convex hull, and $\mathbb{B}$ denotes the closed unit ball in a Euclidean space.

*Then, for every compact set $K \subset \mathbb{R}^n$, every $\epsilon > 0$, and every time horizon $T \in \mathbb{R}_{\geq 0}$ there exists $\eta^\star > 0$ such that: for any $\eta \in (0, \eta^\star]$ and any discrete solution $x_k$ with $x_k(0) \in \bar{K} + \delta\mathbb{B}$, $\delta > 0$, there exists a continuous solution $x_t$ with $x_t(0) \in K$ such that $x_k$ and $x_t$ are $(T, \epsilon)$-close.*

To prove Theorem 2 we will use the results of Theorem 7, where we will have to check that condition (49) is satisfied for forward Euler discretization.

We are now ready to prove Theorem 2. First, note that outer semicontinuity follows from the upper semicontinuity and the closedness of the Filippov differential inclusion map. Furthermore, local boundedness follows from continuity everywhere outside stationary points, which are isolated.

Now, let us examine their discretization by forward-Euler.

The mapping $\mathcal{F}_{\mathrm{d}}$ in this case is a singleton, given by

$$\mathcal{F}_{\mathrm{d}}(x) \triangleq x + \eta F(x), \tag{50}$$

where $\eta > 0$, which clearly satisfies condition (49).

Then, using Theorem 7 we conclude about the $(T, \epsilon)$-closeness between the continuous-time solutions of the flows $\mathcal{F}$ : $q$-RGF (2), $q$-SGF (3), and the discrete-time solutions.

Finally, using the Lyapunov function $V = f - f(x^\star)$ as defined in the proof of Theorem 1, together with inequalities (46g), (35), and a local Lipschitz bound on $f$, one can derive the weak convergence bound given by (16), as follows:

$$
\begin{aligned}
&\|f(x_k) - f(x^\star) - (f(x_t) - f(x^\star))\|_2 = \|f(x_k) - f(x_t)\|_2 \leq \tilde{\epsilon} = L_f \epsilon, \ L_f > 0, \ \epsilon > 0, \\
&\|f(x_k) - f(x^\star)\|_2 - \|(f(x_t) - f(x^\star))\|_2 \leq \|f(x_k) - f(x_t)\|_2 \leq \tilde{\epsilon}, \\
&\|f(x_k) - f(x^\star)\|_2 \leq \tilde{\epsilon} + \|f(x_t) - f(x^\star)\|_2, \\
&\|f(x_k) - f(x^\star)\|_2 \leq \tilde{\epsilon} + [(f(x_0) - f(x^\star))^{(1-\alpha)} - \tilde{c}(1-\alpha)\eta k]^{1/(1-\alpha)}, \text{for } k \leq k^\star,
\end{aligned}
\tag{51}
$$

where $\alpha = \frac{\theta}{\theta'}$, $\theta = \frac{p-1}{\psi}$, $\theta' = \frac{q-1}{q}$, $\tilde{c} = c\left(\left(\frac{\psi}{p-1}\right)^{\frac{p-1}{\psi}} \mu^{\frac{1}{\psi}}\right)^{\frac{1}{\theta'}}$, $k^\star = \frac{(f(x_0)-f(x^\star))^{(1-\alpha)}}{\tilde{c}(1-\alpha)\eta}$.

Next, the case for $k > k^\star$ is rather straightforward: Indeed, for $t > t^\star$ by finite-time convergence result, we have $x_t = x^\star$, which directly leads to the bound $\|f(x_k) - f(x^\star)\|_2 \leq L_f \epsilon, \ k > k^\star$, since the term $\|(f(x_t) - f(x^\star))\|_2$ in (51) vanishes.

## E  PROOF OF THEOREM 3

*Proof.* We divide the discussion by different flows[8].

**Proof of $q$-RGF:**  Following the definitions of smoothness, we have

$$
\begin{aligned}
f(x_{k+1}) &\leq f(x_k) + \langle \nabla f(x_k), x_{k+1} - x_k \rangle + \frac{L}{q}\|x_{k+1} - x_k\|_2^q \\
&= f(x_k) - \gamma \frac{\|\nabla f(x_k)\|_2^2}{\|\nabla f(x_k)\|_2^{\frac{q-2}{q-1}}} + \frac{L\gamma^q}{q}\frac{\|\nabla f(x_k)\|_2^q}{\|\nabla f(x_k)\|_2^{\frac{q(q-2)}{q-1}}} \\
&= f(x_k) - \left(\gamma - \frac{L\gamma^q}{q}\right)\|\nabla f(x_k)\|_2^{\frac{q}{q-1}} \\
&\leq f(x_k) - \frac{q-1}{q}L^{\frac{1}{1-q}}\|\nabla f(x_k)\|_2^{\frac{q}{q-1}} \\
&\leq f(x_k) - \frac{q-1}{q}L^{\frac{1}{1-q}}\frac{q}{q-1}\mu^{\frac{1}{q-1}}(f(x_k) - f^\star) \\
&= f(x_k) - \kappa^{\frac{1}{1-q}}(f(x_k) - f^\star)
\end{aligned}
\tag{52}
$$

---

[8]Note that here for convenience, we fix the order in gradient dominance as $q$, while previous discussion assumes that $q > p$.

so subtract both sides by $f^\star$, we have

$$f(x_{k+1}) - f^\star \leq \left(1 - \kappa^{\frac{1}{1-q}}\right)(f(x_k) - f^\star), \tag{53}$$

which verifies the conclusion by recursion.

**Proof of $q$-SGF:**   Similarly, for the $q$-SGF case, we have

$$
\begin{aligned}
f(x_{k+1}) &\leq f(x_k) + \langle \nabla f(x_k), x_{k+1} - x_k \rangle + \frac{L}{q}\|x_{k+1} - x_k\|_2^q \\
&= f(x_k) - \gamma\|\nabla f(x_k)\|_1^{\frac{1}{q-1}} \langle \nabla f(x_k), \text{sign}\left(\nabla f(x_k)\right) \rangle + \frac{L\gamma^q n^{\frac{q}{2}}}{q}\|\nabla f(x_k)\|_1^{\frac{q}{q-1}} \\
&= f(x_k) - \left(\gamma - \frac{n^{\frac{q}{2}}L\gamma^q}{q}\right)\|\nabla f(x_k)\|_1^{\frac{q}{q-1}} \\
&\leq f(x_k) - \frac{q-1}{q}(n^{\frac{q}{2}}L)^{\frac{1}{1-q}}\|\nabla f(x_k)\|_1^{\frac{q}{q-1}} \\
&\leq f(x_k) - \frac{q-1}{q}(n^{\frac{q}{2}}L)^{\frac{1}{1-q}}\frac{q}{q-1}\mu^{\frac{1}{q-1}}(f(x_k) - f^\star),
\end{aligned}
\tag{54}
$$

so we have

$$f(x_{k+1}) - f^\star \leq \left(1 - (n^{\frac{q}{2}}\kappa)^{\frac{1}{1-q}}\right)(f(x_k) - f^\star). \tag{55}$$

By the inequality $e^{-x} \geq 1 - x$, we can get that the complexity of the above algorithm with $q$-RGF and $q$-SGF are

$$\mathcal{O}\left(\kappa^{\frac{1}{q-1}}\ln\frac{\Delta}{\epsilon}\right) \quad \text{and} \quad \mathcal{O}\left(\left(n^{\frac{q}{2}}\kappa\right)^{\frac{1}{q-1}}\ln\frac{\Delta}{\epsilon}\right), \tag{56}$$

which verifies the conclusion. ∎

## F   PROOF OF STOCHASTIC $q$-RGF (THEOREM 4)

*Proof.* Following the definitions of smoothness, we have

$$
\begin{aligned}
f(x_{k+1}) &\leq f(x_k) + \langle \nabla f(x_k), x_{k+1} - x_k \rangle + \frac{L}{q}\|x_{k+1} - x_k\|_2^q \\
&= f(x_k) - \gamma_k \left\langle \nabla f(x_k) - g(x_k) + g(x_k), \frac{g(x_k)}{\|g(x_k)\|_2^{\frac{q-2}{q-1}}} \right\rangle + \frac{L\gamma_k^q}{q}\frac{\|g(x_k)\|_2^q}{\|g(x_k)\|_2^{\frac{q-2}{q-1}}} \\
&= f(x_k) - \left(\gamma_k - \frac{L\gamma_k^q}{q}\right)\|g(x_k)\|_2^\psi - \gamma_k \left\langle \nabla f(x_k) - g(x_k), \frac{g(x_k)}{\|g(x_k)\|_2^{\frac{q-2}{q-1}}} \right\rangle \\
&\leq f(x_k) - \left(\gamma_k - \frac{L\gamma_k^q}{q}\right)\|g(x_k)\|_2^\psi + \gamma_k \left(\frac{1}{q}\left\|\frac{g(x_k)}{\|g(x_k)\|_2^{\frac{q-2}{q-1}}}\right\|_2^q + \frac{1}{\psi}\|\nabla f(x_k) - g(x_k)\|_2^\psi\right) \\
&= f(x_k) - \left(\frac{\gamma_k}{\psi} - \frac{L\gamma_k^q}{q}\right)\|g(x_k)\|_2^\psi + \frac{\gamma_k}{\psi}\|\nabla f(x_k) - g(x_k)\|_2^\psi,
\end{aligned}
\tag{57}
$$

where the second inequality comes from the Young's inequality. Then note that by Jensen's inequality,

$$
\begin{aligned}
\|\nabla f(x_k)\|_2^\psi &= 2^\psi\left\|\frac{1}{2}(\nabla f(x_k) - g(x_k) + g(x_k))\right\|_2^\psi \leq 2^{\psi-1}\left(\|g(x_k)\|_2^\psi + \|\nabla f(x_k) - g(x_k)\|_2^\psi\right) \\
\implies \quad &-\|g(x_k)\|_2^\psi \leq -2^{1-\psi}\|\nabla f(x_k)\|_2^\psi + \|\nabla f(x_k) - g(x_k)\|_2^\psi
\end{aligned}
\tag{58}
$$

note that the setting of $\gamma_k$ will ensure the coefficient of $\|g(x_k)\|_2^\psi$ to be negative, so we have

$$
f(x_{k+1}) \leq f(x_k) - \left(\frac{\gamma_k}{\psi} - \frac{L\gamma_k^q}{q}\right)\left(2^{1-\psi}\|\nabla f(x_k)\|_2^\psi - \|\nabla f(x_k) - g(x_k)\|_2^\psi\right) + \frac{\gamma_k}{\psi}\|\nabla f(x_k) - g(x_k)\|_2^\psi
$$
$$
= f(x_k) - 2^{1-\psi}\left(\frac{\gamma_k}{\psi} - \frac{L\gamma_k^q}{q}\right)\|\nabla f(x_k)\|_2^\psi + \left(\frac{2\gamma_k}{\psi} - \frac{L\gamma_k^q}{q}\right)\|\nabla f(x_k) - g(x_k)\|_2^\psi,
$$
(59)

then take expectation on both sides, by the bounded variance and gradient dominance, we have

$$
\mathbb{E}f(x_{k+1}) \leq \mathbb{E}\left[f(x_k) - 2^{1-\psi}\left(\frac{\gamma_k}{\psi} - \frac{L\gamma_k^q}{q}\right)\|\nabla f(x_k)\|_2^\psi + \left(\frac{2\gamma_k}{\psi} - \frac{L\gamma_k^q}{q}\right)\left(\frac{\sigma^2}{b(x_k)}\right)^{\frac{\psi}{2}}\right]
$$
$$
\leq \mathbb{E}\left[f(x_k) - 2^{1-\psi}\left(\frac{\gamma_k}{\psi} - \frac{L\gamma_k^q}{q}\right)p\mu^{\frac{1}{q-1}}(f(x_k) - f^\star) + \left(\frac{2\gamma_k}{\psi} - \frac{L\gamma_k^q}{q}\right)\left(\frac{\sigma^2}{b(x_k)}\right)^{\frac{\psi}{2}}\right]
$$
(60)

so subtract both sides by $f^\star$, we have

$$
\mathbb{E}[f(x_{k+1}) - f^\star] \leq \mathbb{E}\left[\left(1 - 2^{1-\psi}\left(\frac{\gamma_k}{\psi} - \frac{L\gamma_k^q}{q}\right)p\mu^{\frac{1}{q-1}}\right)(f(x_k) - f^\star) + \left(\frac{2\gamma_k}{\psi} - \frac{L\gamma_k^q}{q}\right)\left(\frac{\sigma^2}{b(x_k)}\right)^{\frac{\psi}{2}}\right],
$$
(61)

set $\gamma_k \equiv (pL)^{\frac{1}{1-q}}$, $b(x_k) \equiv b^2 = \left(\frac{2\cdot(2\sigma)^\psi \mu^{\frac{1}{1-q}}}{\epsilon}\right)^{\frac{2}{\psi}}$ and

$$
K = \left\lceil \frac{(2\psi)^\psi}{2\kappa^{\frac{1}{1-q}}}\log\left(\frac{2\Delta}{\epsilon}\right)\right\rceil,
$$

we have

$$
\mathbb{E}[f(x_K) - f^\star] \leq \mathbb{E}\left[\left(1 - \frac{2}{(2\psi)^\psi}\kappa^{\frac{1}{1-q}}\right)(f(x_k) - f^\star) + \frac{2\sigma^\psi}{\psi^\psi b^\psi}L^{\frac{1}{1-q}}\right]
$$
$$
\leq \mathbb{E}\left[\left(1 - \frac{2}{(2\psi)^\psi}\kappa^{\frac{1}{1-q}}\right)^K \Delta + \frac{2\sigma^\psi}{\psi^\psi b^\psi}L^{\frac{1}{1-q}}\sum_{i=0}^{K-1}\left(1 - \frac{2}{(2\psi)^\psi}\kappa^{\frac{1}{1-q}}\right)^i\right]
$$
$$
\leq \mathbb{E}\left[\left(1 - \frac{2}{(2\psi)^\psi}\kappa^{\frac{1}{1-q}}\right)^K \Delta + \frac{2\sigma^\psi}{\psi^\psi b^\psi}L^{\frac{1}{1-q}}\frac{1}{1 - \left(1 - \frac{2}{(2\psi)^\psi}\kappa^{\frac{1}{1-q}}\right)}\right]
$$
$$
= \mathbb{E}\left[\left(1 - \frac{2}{(2\psi)^\psi}\kappa^{\frac{1}{1-q}}\right)^K \Delta + \frac{2\sigma^\psi}{\psi^\psi b^\psi}L^{\frac{1}{1-q}}\cdot\frac{(2\psi)^\psi}{2}\kappa^{\frac{1}{q-1}}\right]
$$
(62)
$$
= \mathbb{E}\left[\left(1 - \frac{2}{(2\psi)^\psi}\kappa^{\frac{1}{1-q}}\right)^K \Delta + \frac{(2\sigma)^\psi \mu^{\frac{1}{1-q}}}{b^\psi}\right]
$$
$$
\leq \mathbb{E}\left[\exp\left(-\frac{2\kappa^{\frac{1}{1-q}}}{(2\psi)^\psi}K\right)\Delta + \frac{(2\sigma)^\psi \mu^{\frac{1}{1-q}}}{b^\psi}\right]
$$
$$
\leq \frac{\epsilon}{2} + \frac{\epsilon}{2} = \epsilon,
$$

while the total sample complexity is (note that $\frac{1}{\psi} + \frac{1}{q} = 1$, $q \in (1,2]$, which implies $\psi \in [2, +\infty)$)

$$
K \cdot b^2 \geq \frac{(2\psi)^\psi}{2\kappa^{\frac{1}{1-q}}}\log\left(\frac{2\Delta}{\epsilon}\right)\cdot\left(\frac{2\cdot(2\sigma)^\psi \mu^{\frac{1}{1-q}}}{\epsilon}\right)^{\frac{2}{\psi}} = \mathcal{O}\left(\sigma^2\kappa^{\frac{1}{q-1}}\mu^{-\frac{2}{q}}\psi^\psi\epsilon^{-\frac{2}{\psi}}\log\frac{\Delta}{\epsilon}\right),
$$
(63)

which verifies the conclusion. ∎

# G   PROOF OF STOCHASTIC $q$-SGF (THEOREM 5)

*Proof.* Note that, as mentioned in Assumption 4 and 5, here the gradient dominance and Lipschitz smoothness is defined under $(\ell_1, \ell_\infty)$-norms. Following the definitions of smoothness, we have (note that $\psi = \frac{q}{q-1}$)

$$
\begin{aligned}
f(x_{k+1}) &\leq f(x_k) + \langle \nabla f(x_k), x_{k+1} - x_k \rangle + \frac{L}{q} \|x_{k+1} - x_k\|_\infty^q \\
&= f(x_k) - \gamma_k \|g(x_k)\|_1^{\frac{1}{q-1}} \langle \nabla f(x_k), \operatorname{sign}(g(x_k)) \rangle + \frac{L\gamma_k^q}{q} \|g(x_k)\|_1^{\frac{q}{q-1}} \cdot \|\operatorname{sign}(g(x_k))\|_\infty^q \\
&= f(x_k) - \gamma_k \|g(x_k)\|_1^{\frac{1}{q-1}} \langle \nabla f(x_k), \operatorname{sign}(g(x_k)) \rangle + \frac{L\gamma_k^q}{q} \|g(x_k)\|_1^{\frac{q}{q-1}},
\end{aligned}
\tag{64}
$$

conditioning on $x_k$, we have

$$
\begin{aligned}
&\mathbb{E}[f(x_{k+1}) \mid x_k] \\
&\leq f(x_k) - \mathbb{E}\left[ \gamma_k \|g(x_k)\|_1^{\frac{1}{q-1}} \langle \nabla f(x_k), \operatorname{sign}(g(x_k)) \rangle \mid x_k \right] + \mathbb{E}\left[ \frac{L\gamma_k^q}{q} \|g(x_k)\|_1^{\frac{q}{q-1}} \mid x_k \right],
\end{aligned}
\tag{65}
$$

note that for each component of $g(x_k)$, we have

$$
\begin{aligned}
&\mathbb{E}\left[ \|g(x_k)\|_1^{\frac{1}{q-1}} \operatorname{sign}(g_i(x_k)) \mid x_k \right] \\
&= \mathbb{E}\Big[ \|g(x_k)\|_1^{\frac{1}{q-1}} \mathbb{P}(\operatorname{sign}(\nabla_i f(x_k)) = \operatorname{sign}(g_i(x_k))) \operatorname{sign}(\nabla_i f(x_k)) \\
&\qquad\qquad - \|g(x_k)\|_1^{\frac{1}{q-1}} \mathbb{P}(\operatorname{sign}(\nabla_i f(x_k)) \neq \operatorname{sign}(g_i(x_k))) \operatorname{sign}(\nabla_i f(x_k)) \mid x_k \Big] \\
&= \mathbb{E}\left[ \|g(x_k)\|_1^{\frac{1}{q-1}} (2\mathbb{P}(\operatorname{sign}(\nabla_i f(x_k)) = \operatorname{sign}(g_i(x_k))) - 1) \operatorname{sign}(\nabla_i f(x_k)) \mid x_k \right],
\end{aligned}
\tag{66}
$$

then with the SPB condition in Assumption 5, we have

$$
\mathbb{P}(\operatorname{sign}(\nabla_i f(x_k)) = \operatorname{sign}(g_i(x_k))) \geq p^* > \frac{1}{2},
$$

so we have

$$
\begin{aligned}
&\mathbb{E}[f(x_{k+1}) \mid x_k] \\
&\leq f(x_k) - \mathbb{E}\left[ \gamma_k \|g(x_k)\|_1^{\frac{1}{q-1}} (2p^* - 1) \langle \operatorname{sign}(\nabla f(x_k)), \nabla f(x_k) \rangle \mid x_k \right] + \mathbb{E}\left[ \frac{L\gamma_k^q}{q} \|g(x_k)\|_1^{\frac{q}{q-1}} \mid x_k \right] \\
&\leq f(x_k) - \mathbb{E}\left[ \gamma_k (2p^* - 1) \|g(x_k)\|_1^{\frac{1}{q-1}} \|\nabla f(x_k)\|_1 \mid x_k \right] + \mathbb{E}\left[ \frac{L\gamma_k^q}{q} \|g(x_k)\|_1^{\frac{q}{q-1}} \mid x_k \right] \\
&\leq f(x_k) - \gamma_k (2p^* - 1) \|\nabla f(x_k)\|_1^{\frac{q}{q-1}} + \frac{L\gamma_k^q}{q} \mathbb{E}\left[ \|g(x_k)\|_1^{\frac{q}{q-1}} \mid x_k \right]
\end{aligned}
\tag{67}
$$

where the last inequality comes from the convexity of $\|\cdot\|_1^{\frac{1}{q-1}}$. Then for the last term in the RHS, by decomposition, we have

$$
\|g(x_k)\|_1^\psi = 2^\psi \left\| \frac{1}{2}(\nabla g(x_k) - f(x_k) + f(x_k)) \right\|_1^\psi \leq 2^{\psi-1}\left( \|f(x_k)\|_1^\psi + \|\nabla f(x_k) - g(x_k)\|_1^\psi \right),
\tag{68}
$$

so we have (recall that $\psi = \frac{q}{q-1}$)

$$\mathbb{E}[f(x_{k+1}) \,|\, x_k]$$

$$\leq f(x_k) - \gamma_k(2p^* - 1)\|\nabla f(x_k)\|_1^{\frac{q}{q-1}} + \frac{L\gamma_k^q 2^{\psi-1}}{q}\mathbb{E}\Big[\|f(x_k)\|_1^{\psi} + \|\nabla f(x_k) - g(x_k)\|_1^{\psi} \,|\, x_k\Big]$$

$$\leq f(x_k) - \left((2p^* - 1)\gamma_k - \frac{L\gamma_k^q 2^{\psi-1}}{q}\right)\|\nabla f(x_k)\|_1^{\frac{q}{q-1}} + \frac{L\gamma_k^q 2^{\psi-1}}{q}\left(\frac{\sigma^2}{b(x_k)}\right)^{\frac{\psi}{2}}$$

$$\leq f(x_k) - \left((2p^* - 1)\gamma_k - \frac{L\gamma_k^q 2^{\psi-1}}{q}\right)\frac{q\mu^{\frac{1}{q-1}}}{q-1}(f(x_k) - f^\star) + \frac{L\gamma_k^q 2^{\psi-1}}{q}\left(\frac{\sigma^2}{b(x_k)}\right)^{\frac{\psi}{2}},$$

$$(69)$$

so take

$$\gamma_k \equiv \left(\frac{L2^{\psi-1}}{2p^* - 1}\right)^{\frac{1}{1-q}}, \quad b(x_k) \equiv b^2 = \left(\frac{2\mu^{\frac{1}{1-q}}}{q}\right)^{\frac{2}{\psi}}(2p^* - 1)^{-4}\sigma^2\epsilon^{-\frac{2}{\psi}},$$

and

$$K = \left\lceil (2p^* - 1)^{-\psi}\big(\kappa 2^{\psi-1}\big)^{\frac{1}{q-1}}\ln\frac{2\Delta}{\epsilon}\right\rceil,$$

we have

$$\mathbb{E}[f(x_K) - f^\star]$$

$$\leq \mathbb{E}\left[\left(1 - \frac{1}{\psi}(2p^* - 1)^{\psi}\big(L2^{\psi-1}\big)^{\frac{1}{1-q}}\frac{q\mu^{\frac{1}{q-1}}}{q-1}\right)(f(x_{K-1}) - f^\star) + \frac{\big(L2^{\psi-1}\big)^{\frac{1}{1-q}}}{q(2p^* - 1)^{\psi}}\left(\frac{\sigma^2}{b^2}\right)^{\frac{\psi}{2}}\right]$$

$$= \mathbb{E}\left[\left(1 - (2p^* - 1)^{\psi}\big(\kappa 2^{\psi-1}\big)^{\frac{1}{1-q}}\right)(f(x_{K-1}) - f^\star) + \frac{\big(L2^{\psi-1}\big)^{\frac{1}{1-q}}}{q(2p^* - 1)^{\psi}}\left(\frac{\sigma^2}{b^2}\right)^{\frac{\psi}{2}}\right]$$

$$\leq \left(1 - (2p^* - 1)^{\psi}\big(\kappa 2^{\psi-1}\big)^{\frac{1}{1-q}}\right)^K(f(x_0) - f^\star) + \frac{\big(L2^{\psi-1}\big)^{\frac{1}{1-q}}}{q(2p^* - 1)^{\psi}}\left(\frac{\sigma^2}{b^2}\right)^{\frac{\psi}{2}} \cdot \frac{1}{(2p^* - 1)^{\psi}\big(\kappa 2^{\psi-1}\big)^{\frac{1}{1-q}}}$$

$$\leq \exp\left(-(2p^* - 1)^{\psi}\big(\kappa 2^{\psi-1}\big)^{\frac{1}{1-q}}K\right)\Delta + \frac{(\mu)^{\frac{1}{1-q}}}{q(2p^* - 1)^{2\psi}}\left(\frac{\sigma^2}{b^2}\right)^{\frac{\psi}{2}}$$

$$= \frac{\epsilon}{2} + \frac{\epsilon}{2} = \epsilon,$$

$$(70)$$

which concludes the proof[9]. ∎

**Remark 6.** Similar results can be derived in the case of single sample (trivial batch size), however, some of them will require additional assumptions like bounded gradient norm and vanishing stepsize (Karimi et al. (2016), Theorem 4), but here we do not need this strong assumption of gradient boundedness, and we always use a constant stepsize; these are the benefits brought by 'non-trivial' batch-size, i.e., neither one sample nor the full set of data.

Furthermore, in that case, the final complexity will be generally $\mathcal{O}(1/\epsilon)$ (Karimi et al. (2016), Theorem 4), whereas here our result is a linear convergence iteration complexity (i.e., $\mathcal{O}(\log(1/\epsilon))$) to achieve the $\epsilon$-neighborhood (while the final sample complexities of two paradigms should be the same), which is much faster. So there will be a tradeoff in different parameter settings.

Lastly, we want to mention that batch size with dependence on $\epsilon$ is pretty common (or even required) in stochastic optimization problems, e.g., Ghadimi et al. (2016) and Fang et al. (2018). Therefore, we believe that using a batch size with dependence on $\epsilon$ is a reasonable option.

---

[9]Different from stochastic RGF, here we do not discuss the sample complexity, because the extra SPB assumption is defined on the "oracle", i.e., the mini-batch estimator $g(x)$, so the concern here should be the oracle complexity instead, which will be the iteration number $K$ here.

## H  ADDITIONAL EXPERIMENT DETAILS AND NUMERICAL RESULTS

In this section, we will expand upon the numerical results experiments discussed in the paper. In particular, we report more details on the hyper-parameters values used[10] for the numerical tests, and report some results for GPU implementation of SVHN experiments.

### H.1  HYPER PARAMETERS VALUES USED IN THE TESTS OF SECTION 4.1- EXAMPLE 2

- GD fixed step size: $\eta = 10^{-3}$
- RGF Euler disc. w/fixed step size: $q = 2.2$, $\eta = 10^{-3}$
- RGF Euler disc. w/fixed step size: $q = 3$, $\eta = 10^{-2}$
- RGF Euler disc. w/fixed step size: $q = 6$, $\eta = 10^{-2}$
- RGF Euler disc. w/fixed step size: $q = 10$, $\eta = 10^{-2}$

### H.2  HYPER PARAMETERS VALUES USED IN THE TESTS OF SECTION 4.2

Note that the description of the coefficients for each of the prior art methods can be found in: https://pytorch.org/docs/stable/optim.html.

- GD: $\eta = 4.10^{-2}$, $\mu = 0.9$, Nesterov=False
- RGF: $\eta = 4.10^{-2}$
- SGF: $\eta = 4.10^{-3}$
- ADAM: $\eta = 8.10^{-4}$ (remaining coefficients=nominal values)
- RMS: $\eta = 10^{-3}$ (remaining coefficients=nominal values)
- ADAGRAD: $\eta = 10^{-3}$ (remaining coefficients=nominal values)
- ADADELTA: $\eta = 4.10^{-2}$, $\rho = 0.9$, $\epsilon = 10^{-6}$, weight decay $= 0$

**Remark 7. Choice of $q$:** The settling time upper bound (15) decreases as $q \to \infty$, which appears to lead to faster convergence when discretized. On the other hand, the larger $q$ is, the stiffer the ODE, so more prone to numerical instability, so $q$ cannot be too large. Therefore, assuming $p$ to be not too large, it appears that $q \in (p, p + \delta]$ works best, with $\delta > 0$ as small as needed to avoid numerical issues. For instance, if we know the cost function to be strongly convex (locally), then we search for $q$ slightly larger than $p = 2$ at first, but continue to increase until performance deteriorates. If, on the other hand, we don't know the order $p > 1$, then it's currently unclear how to choose $q$. We will investigate this further in future work. Furthermore, there is evidence that gradient dominance does hold locally in many deep learning contexts (Zhou and Liang, 2017, https://arxiv.org/abs/1710.06910). Indeed, since convexity readily leads to gradient dominance of order $p = \infty$, it suffices that a slightly stronger form of it holds (but weaker than strong convexity), in order to have $p < \infty$, and thus for us to be able to choose $q > p$.

### H.3  EXPERIMENT 6: GPU IMPLEMENTATION

To check if the numerical results obtained on CPU, more specifically the acceleration trends, hold true on a GPU, we run some extra tests on a Titan X nvidia gpu with 12GB memory, the results are reported below.

We first tested the performance of Euler discretization of the proposed flows against Adam, and GD algorithms on SVHN dataset. We tested the proposed algorithms to train the VGG16 CNN model with cross entropy loss. We divided the dataset into a training set of 74 batches with 1000 images each, and a test set of 27 batches of 1000 images each, and ran 20 epochs of training over all the training batches. We tested Euler discretization of $q$-RGF ($c = 1$, $q = 2.1$, $\eta = 0.04$ ), and Euler discretization of $q$-SGF ($c = 10^{-3}$, $q = 2.1$, $\eta = 0.04$ ) against GD ($\eta = 0.1$) and Adam (same optimal tuning as in Section 4.2).

In Figures 5 , 6 we can see that both algorithms, Euler $q$-RGF and Euler $q$-SGF, converge faster than GD and Adam for these tests, and reach the same performance on the test-set.

---

[10]In all the tests, for $q$-RGF and $q$-SGF $c = 1$ unless otherwise stated.

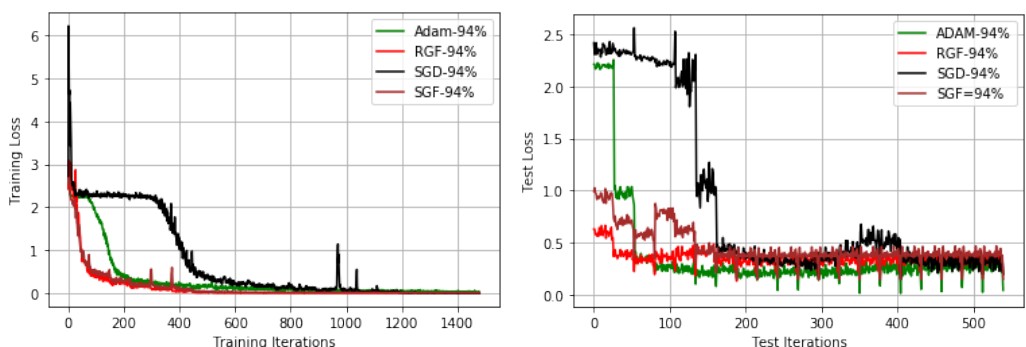

Figure 5: Losses for several optimization algorithms run on GPU- VGG16-SVHN: Train loss (left), test loss (right)

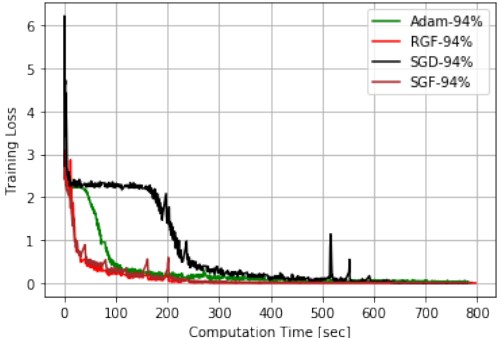

Figure 6: Training loss vs. computation time for several optimization algorithms run on GPU-VGG16-SVHN

