# OpenReview forum: "First-Order Optimization Inspired from Finite-Time Convergent Flows"
_ICLR.cc/2022/Conference — ICLR 2022 Submitted_

### Official Review · Reviewer_5qiS · 2021-10-30

**Correctness:** 3
**Technical Novelty And Significance:** 2
**Empirical Novelty And Significance:** 2
**Recommendation:** 6
**Confidence:** 4

**Main Review:**

My comments are as follows.

1) The paper starts well but in Eq. (6a)-(6b), when authors describe the state-space form of the optimization algorithms, I think there might be a typo in (6b): Is it supposed to be $G(X_{k+1})$? Thiis notation is confusing as it is not made clear what $x_k$ and $X_k$ actually denote.

2) The following examples are meant to be helpful but they all consider the case $G(x) = x$. It is really unclear what (6a)-(6b) describes. Authors should put an example with a clear case where $G$ is not identity (e.g. a proximal-gradient scheme?)

I also don’t understand the introduction of this notation, as it looks like it’s not used elsewhere in the paper and not central.

3) The authors then introduce the fundamental assumption of the cost functions considered, that is, gradient dominance of order $p$. I strongly suggest authors to motivate this assumption beyond strong convexity. What kind of non-convex functions does satisfy this inequality? Is it similar to, e.g., weak-convexity or dissipativity conditions usually used in non-convex optimization literature?

4) The toy examples in the paper are helpful but their connection to theory is not exploited: In particular, in this section, I’d strongly suggest authors to verify their assumptions for these toy examples precisely by working out relevant constants. Then, plotting the errors vs. actual error bound computations would really be insightful about the meaning of the results derived in this paper.

**Summary Of The Paper:**

This paper considers the analysis of two discrete time schemes derived from gradient flows named q-RGF and q-SGF. The topic fits more generally into continuous time perspectives for optimization and the relations between ODE theory and optimization. This is an interesting direction with many interesting promises.

**Summary Of The Review:**

These results can be potentially useful for advances in this field. I found the paper clear and insightful in general.

---

> ### Author Response · Authors · 2021-11-15
> **Response to Reviewer 5qiS**
>
> Thank you very much for your time and we appreciate your review. Please check our response to your review.
>
> ---
>
> **Problem 1**: *In Eq. (6a)-(6b), when authors describe the state-space form of the optimization algorithms, I think there might be a typo in (6b): Is it supposed to be $G(X_{k+1})$? This notation is confusing as it is not made clear what $x_k$ and $X_k$ actually denote.*
>
> **Response 1**: Thank you for the question. We agree, and apologize for the confusing presentation in Example 1. The goal was to give a general presentation of optimization algorithms as a discrete dynamical system, with internal states $X_k$ that include $x$ but might also include other variables, e.g., momentum variables in higher order methods, or simply time-varying step size. We have removed all the statement pertaining to feedback control, and made the statement more concise and clear.
>
> ---
>
> **Problem 2**: *The following examples are meant to be helpful but they all consider the case $G(x)=x$. It is really unclear what (6a)-(6b) describes. Authors should put an example with a clear case where $G$ is not identity (e.g. a proximal-gradient scheme?)*
>
>
> **Response 2**: We agree, and as indicated above, we have corrected the typos in the formulation of (6), i.e., $F_d(k,X_k)\in R^{m}$ not $R^{n}$ as it was stated. We also removed the confusing statements about feedbacks, and simply shown that the case with time-varying step-size $\eta_k$, can be formulated as in (6), where the extended vector $X_k$ is defined as $X_k=[\eta_k,x_k]$, and where part of $F_d$, is defined by the update of $\eta_k$, i.e., $\eta_{k+1}=F_d (\eta_k,x_k)$. Other momentum-based method can be written in the form of (6), where the extended vector $X_k$ will contain $x_k$, as well as the momentum variables.
>
> ---
>
> **Problem 3**: *The authors then introduce the fundamental assumption of the cost functions considered, that is, gradient dominance of order $p$. I strongly suggest authors to motivate this assumption beyond strong convexity. What kind of non-convex functions does satisfy this inequality? Is it similar to, e.g., weak-convexity or dissipativity conditions usually used in non-convex optimization literature?*
>
>
> **Response 3**: Thank you for indicating this important point. For the generalized gradient dominance condition, this is a very important assumption in nonconvex optimization (of course there are many other works further relaxing the assumption or using other settings), also there are some other conditions like weak convexity (which are very different). We agree that this assumption is not that weak, while we also want to emphasize that it is in fact widely verified in practice, e.g., in Mei et al. (2021), showed that in reinforcement learning with softmax policy parameterization, the objective function can satisfy \L{}ojasiewicz inequality (or KL inequality (Kurdyka, 1998)), which is just the gradient dominance condition here; also in deep learning there is evidence that gradient dominance does hold locally, e.g., Zhou and Liang, (2017). So we believe that our setting is still reasonable. We added these references in the revised version of the paper to motivate the gradient dominance assumption.
>
> ---
>
> **Problem 4**: *The toy examples in the paper are helpful but their connection to theory is not exploited: In particular, in this section, I’d strongly suggest authors to verify their assumptions for these toy examples precisely by working out relevant constants. Then, plotting the errors vs. actual error bound computations would really be insightful about the meaning of the results derived in this paper.*
>
>
> **Response 4**: Thank you very much for this interesting suggestion. In fact as mentioned in Lei and Ying, (2020), our example is a special case of the so-called $q$-th power absolute distance loss, and it will be Lipschitz smooth with of order $q$ when $q\in (0,1]$, so the proposed example in fact satisfy the requirement. Following your suggestion, we have revised the description of this part in the submission and updated the experiment result with the theoretical bound from Theorem 3.
>
> ---
>
> References:
>
> [1] Mei, Jincheng, et al. "Leveraging non-uniformity in first-order non-convex optimization." arXiv preprint arXiv:2105.06072 (2021).
>
> [2] Zhou, Yi, and Yingbin Liang. "Characterization of gradient dominance and regularity conditions for neural networks." arXiv preprint arXiv:1710.06910 (2017).
>
> [3] Kurdyka, Krzysztof. "On gradients of functions definable in o-minimal structures." Annales de l'institut Fourier. Vol. 48. No. 3. 1998.
>
> [4] Lei, Yunwen, and Yiming Ying. "Fine-grained analysis of stability and generalization for stochastic gradient descent." International Conference on Machine Learning. PMLR, 2020.

---

> ### Author Response · Authors · 2021-11-29
> **Thank you, did our response address your concerns?**
>
> Dear Reviewer 5qiS,
>
> We appreciate your opinions in our work. We have responded in the rebuttal and revised the submission, also we revised the plot of Experiment 1 per your request, trying to fully address your concerns in the review.
>
> We would really appreciate if you could read our replies and let us know if your main concerns have been addressed, or if you have anymore questions for us. Alternatively, if you found our responses to be satisfactory, we would appreciate if you could revise your scores accordingly. Thank you very much.
>
> Best regards,
>
> Authors

---

### Official Review · Reviewer_xRJ7 · 2021-10-31

**Correctness:** 4
**Technical Novelty And Significance:** 2
**Empirical Novelty And Significance:** Not applicable
**Recommendation:** 5
**Confidence:** 3

**Main Review:**

Strength:
1. Convergence behavior of the discretization of gradient flows is an important question in optimization.
2. The paper is well written and very good to follow.
3. The methodology is validated via empirical experiments and theoretical analysis.

Weakness:
1. The convergence rate of Euler discretizations of rescaled gradient flow and signed gradient flow under the deterministic setting exists in the literature. (The author do point out this.) The only difference is the generalization of the Lipschitz smoothness. The author should illustrate why such generalization is important and what is the main technical difficulty in extending the existing theoretical result.
2. Under the stochastic setting, the result is not convincing in the sense that although the epsilon can be arbitrary small, the batch size will go to infinity, which is not good especially under the stochastic setting.

**Summary Of The Paper:**

In this paper, the author study two first-order optimization methods, which are the Euler discretizations of rescaled gradient flow and signed gradient flow. They prove the convergence rates for both methods under both deterministic and stochastic settings.

**Summary Of The Review:**

I found the manuscript to be clearly written and technically sound. I am, rather unfortunately, inclined to recommend that the manuscript be rejected on the grounds that the work may not have sufficient merit / novelty to warrant publication.

---

> ### Author Response · Authors · 2021-11-15
> **Response to Reviewer xRJ7**
>
> Thank you very much for your time and we appreciate your review. Please check our response to your review.
>
> ---
>
> **Problem 1**: *The convergence rate of Euler discretizations of rescaled gradient flow and signed gradient flow under the deterministic setting exists in the literature. (The author do point out this.) The only difference is the generalization of the Lipschitz smoothness. The author should illustrate why such generalization is important and what is the main technical difficulty in extending the existing theoretical result.*
>
>
> **Response 1**: Thank you for this question, here we will further illustrate our motivation.
>
> As mentioned in Remark 3, we generalize the standard gradient flow and sign-based algorithm to a more generalized setting with Holder smoothness, also the generalized gradient dominance condition.
>
> We want to argue that the assumptions are generalized to fit more problem settings. First for the Holder smoothness, we want to argue that it is an important generalization of Lipschitz smoothness, it has been shown that there are some applications that satisfying Holder smoothness while not satisying the classical Lipschitz smoothness, e.g., $q$-norm hinge loss and $q$-th power absolute distance loss for regression (Lei and Ying, 2020).
>
> Then for the generalized gradient dominance condition is a very important assumption in nonconvex optimization (of course there are many other works further relaxing the assumption or using other settings). We agree that it is not that weak, but we also want to emphasize that it is in fact widely verified in practice, e.g., in Mei et al. (2021), they showed that in reinforcement learning with softmax policy parameterization, the value function will satisfy \L{}ojasiewicz inequality (or KL inequality (Kurdyka, 1998)), which is just the gradient dominance condition here; also in deep learning there is evidence that gradient dominance does hold locally, e.g., Zhou and Liang, (2017). Furthermore, we want to emphasise that other works who present analysis of rescaled gradient flow, do so under stronger assumptions, e.g., strong smoothness (Wilson et al. (2019), Definition 2), which are based on higher-order information.
>
> In terms of the analysis difficulty, due to the exponent $q$, it will bring huge difference in the error decomposition for convergence rate derivation (compared to the classical square case), especially in the stochastic case, which drives us to derive new bounds on the gradient estimation. Overall, we think the generalization here is important and definitely brings challenges in the analysis.
>
> ---
>
> **Problem 2**: *Under the stochastic setting, the result is not convincing in the sense that although the epsilon can be arbitrary small, the batch size will go to infinity, which is not good especially under the stochastic setting.*
>
> **Response 2**: Thank you for indicating this point. While it is very common in stochastic optimization algorithms, we basically view it as a tradeoff, rather than a weakness. For simplicity, let us use the standard case of $q$-RGF with $q=2$ for illustration.
>
> First for single sample result (batch size = 1), we believe that the corresponding results can be established, while some of them will require additional assumptions like bounded gradient norm and vanishing stepsize (Karimi et al., 2016, Theorem 4), but here we do not need this strong assumption of gradient boundedness, and we always use a constant stepsize; these are the benefits brought by `non-trivial' batch-size, i.e., neither one sample nor the full set of data.
>
> Furthermore, in that case, the final complexity will be generally $\mathcal{O}(1/\epsilon)$ (Karimi et al., 2016, Theorem 4), whereas here our result is a linear convergence iteration complexity (i.e., $\mathcal{O}(\log (1/\epsilon))$) to achieve the $\mathcal{\epsilon}$-neighborhood (while the final sample complexities of two paradigms should be the same), which is much faster. So there will be a tradeoff in different parameter settings.
>
> Lastly, we want to mention that batch size with dependence on $\epsilon$ is pretty common (or even required sometimes) in stochastic optimization problems, e.g., Ghadimi et al. (2016) and Fang et al. (2018). Therefore, we believe that using a batch size with dependence on $\epsilon$ is a reasonable option, while the case of single sample (or trivial batch) can be derived from our setting, under additional stronger gradient boundedness assumptions. We added Remark 6 explaining this trade-off in the revised version of the paper.

---

> > ### Author Response · Authors · 2021-11-15
> > **Response to Reviewer xRJ7 (Part 2)**
> >
> > References:
> >
> > [1] Karimi, Hamed, Julie Nutini, and Mark Schmidt. "Linear convergence of gradient and proximal-gradient methods under the polyak-łojasiewicz condition." Joint European Conference on Machine Learning and Knowledge Discovery in Databases. Springer, Cham, 2016.
> >
> > [2] Ghadimi, Saeed, Guanghui Lan, and Hongchao Zhang. "Mini-batch stochastic approximation methods for nonconvex stochastic composite optimization." Mathematical Programming 155.1-2 (2016): 267-305.
> >
> > [3] Fang, Cong, et al. "SPIDER: near-optimal non-convex optimization via stochastic path integrated differential estimator." Proceedings of the 32nd International Conference on Neural Information Processing Systems. 2018.
> >
> > [4] Mei, Jincheng, et al. "Leveraging non-uniformity in first-order non-convex optimization." arXiv preprint arXiv:2105.06072 (2021).
> >
> > [5] Zhou, Yi, and Yingbin Liang. "Characterization of gradient dominance and regularity conditions for neural networks." arXiv preprint arXiv:1710.06910 (2017).
> >
> > [6] Kurdyka, Krzysztof. "On gradients of functions definable in o-minimal structures." Annales de l'institut Fourier. Vol. 48. No. 3. 1998.
> >
> > [7] Lei, Yunwen, and Yiming Ying. "Fine-grained analysis of stability and generalization for stochastic gradient descent." International Conference on Machine Learning. PMLR, 2020.
> >
> > [8] Wilson, Ashia C., Lester Mackey, and Andre Wibisono. "Accelerating Rescaled Gradient Descent: Fast Optimization of Smooth Functions." Advances in Neural Information Processing Systems 32 (2019): 13555-13565.

---

> ### Author Response · Authors · 2021-11-29
> **Thank you, did our response address your concerns?**
>
> Dear Reviewer xRJ7,
>
> We appreciate your opinions in our work. We have responded in the rebuttal and revised the submission, trying to fully address your concerns in the review.
>
> We would really appreciate if you could read our replies and let us know if your main concerns have been addressed, or if you have anymore questions for us. Alternatively, if you found our responses to be satisfactory, we would appreciate if you could revise your scores accordingly. Thank you very much.
>
> Best regards,
>
> Authors

---

### Official Review · Reviewer_AvzH · 2021-11-01

**Correctness:** 3
**Technical Novelty And Significance:** 2
**Empirical Novelty And Significance:** 2
**Recommendation:** 6
**Confidence:** 4

**Main Review:**

The submitted paper studies the convergence properties of discretized gradient flows, and provides some convergence results which is the main strength of the paper. To me, there are few points which are not clear
1. In Theorem 2, we have $k^\star = \frac{(f(x_0)-f(x^\star))^{(1-\alpha)}}{\tilde{c}(1-\alpha) \eta}$. Now denote $C = \frac{(f(x_0)-f(x^\star))^{(1-\alpha)}}{\tilde{c}(1-\alpha)}$ which is a constant. Suppose $\epsilon$ is some chosen accuracy and $\eta$ is some sufficiently small time step-size. The condition $|t-k\eta|<\epsilon$ means that $$ \frac{t-\epsilon}{\eta} < k < \frac{t+\epsilon}{\eta}  . $$ For $t$ large enough, it can happen that $$ t - \epsilon > C $$ where $C$ is defined above. Corresponding we have $k > k^\star$. Then Equation 16 would fail since $k \leq k^\star$? I'm wondering if I missed something here?
2. The results for stochastic gradient descent are $\epsilon$-dependent, such as the batch size which is $O(\frac{1}{\epsilon})$ for stochastic $q$-RGF and $q$ closed to $2$. So in the numerical experiments, what are the choices of $\epsilon$? And the result is not very practical in the sense that if sufficiently high accuracy solution is desired, then the batch size would be too large such that it will be close or equal to the full gradient descent. As a result, is it possible to establish result for accuracy independent batch size or for the vanishing step-size?

Also I have two side questions
1. Since the discrete algorithms fails the finite-time convergence, is it possible to have other type of discretization which can provide finite convergence? Or at least numerically.
2. Suppose we consider second-order dynamic systems, can the finite-time convergence still hold in this case?



**Summary Of The Paper:**

Motivated from the finite-time convergence of two first-order gradient flows, the submitted paper studied the convergence properties of discrete iteration schemes in both deterministic and stochastic settings. Convergence results are proved for these algorithms, and numerical experiments are provided to demonstrate the performance of the schemes.

**Summary Of The Review:**

Motivated from continuous dynamical system, the authors studied the convergence behaviors of discrete first-order methods. However, there is a gap between the continuous and discrete setting, and the obtained results need to be clarified.


edit: the responses look good to me, and i raised my score by 1.

---

> ### Author Response · Authors · 2021-11-15
> **Response to Reviewer AvzH**
>
> Thank you very much for your time and we appreciate your review. Please check our response to your review.
>
> ---
>
> **Problem 1**: *Then Equation 16 would fail since $k\leq k^*$? I'm wondering if I missed something here?*
>
> **Response 1**: Thank you for the question. We think there is a misunderstanding in the interpretation of Theorem 2. The theorem states the closeness result only for $t\in[0, t^*]$, where $ t^*=k^* \eta $. When we reach $t^*$ the finite-time convergence result in continuous-time guarantees that the state $x(t)$ reaches the equilibrium point $x^*$, and stays at this point (by definition of equilibrium point $\frac{d x(t)}{dt}|_{t=t^*}=0$; see Definition 2 in Appendix B of the paper). This means that in discrete-time after reaching $x^*$ at $k=k^*$, the bound in (16) simplifies to the term $L_f\epsilon$; the $L_f$ is due to the Lipschitz constant of $f$, and $\epsilon$ is due to $x$ being in the $\epsilon$-neighborhood of $x^*$.
>
> We have modified the theorem (and its proof) to explicitly state the bound in (16) for $k>k^*$, i.e., $  \|f(x_k)-f(x^*)\|_2\leq L_f \epsilon,$ for $k>k^*$
>
> ---
>
> **Problem 2**: *The results for stochastic gradient descent are $\epsilon$-dependent...... So in the numerical experiments, what are the choices of $\epsilon$? And the result is not very practical in the sense that if sufficiently high accuracy solution is desired, then the batch size would be too large such that it will be close or equal to the full gradient descent. As a result, is it possible to establish result for accuracy independent batch size or for the vanishing step-size?*
>
>
> **Response 2**: Thank you very much for this interesting problem. For simplicity, let us use the standard case of $q$-RGF with $q=2$ for illustration.
>
> First for single sample result (or "accuracy independent batch size"), we agree with you, we believe that the corresponding results can be established, while some of them will require additional assumptions like bounded gradient norm and vanishing stepsize (Karimi et al., 2016, Theorem 4), but here we do not need this strong assumption of gradient boundedness, and we always use a constant stepsize; these are the benefits brought by `non-trivial' batch-size, i.e., neither one sample nor the full set of data.
>
> Furthermore, in that case, the final complexity will be generally $\mathcal{O}(1/\epsilon)$ (Karimi et al., 2016, Theorem 4), whereas here our result is a linear convergence iteration complexity (i.e., $\mathcal{O}(\log (1/\epsilon))$) to achieve the $\mathcal{\epsilon}$-neighborhood (while the final sample complexities of the two paradigms should be the same), which is much faster. So there will be a tradeoff in different parameter settings.
>
> Last we want to mention that, batch size with dependence on $\epsilon$ is pretty common (or even required sometimes) in stochastic optimization problems, e.g., Ghadimi et al. (2016) and Fang et al. (2018).
>
> In practice, generally the batch size is a common parameter to be tuned, we agree that it is a common gap between theory and practice, a potential explanation is that, the complexity bounds here is the worst-case complexity, while the practical problems may not be the "worst case", so sometimes we can use smaller batch size in practice.
>
> To summarize, we agree with you in the aspect that there will be some tradeoff between mini-batch and single-sample algorithms, and it can be expected that the single-sample result (or "accuracy independent batch size") will be established, we will definitely try to explore it in the future. We added Remark 6 explaining this trade-off in the revised version of the paper.
>
> ---
>
> **Problem 3**: *Since the discrete algorithms fails the finite-time convergence, is it possible to have other type of discretization which can provide finite convergence? Or at least numerically.*
>
> **Response 3**: Thank you for the question. The concept of finite-time convergence in continuous-time does not translate to a similar concept of finite-time convergence in discrete-time. In the sense that we were not expecting to be able to converge in a desired predefined number of discrete steps. All what we are hoping for is to derive some competitive convergence rates after discretizing the continuous flow, and to observe empirical acceleration w.r.t. some existing methods. We agree that explicit forward Euler discretization is one basic explicit method for discretization, i.e., solving the continuous flow, and we agree that other, possibly implicit or of higher order, discretization methods, e.g. Runge-Kutta, would lead to better (in terms of convergence speed) numerical results, since they are designed to make the discrete solutions "stay closer" (than a first order Euler method) to the continuous solutions. We are working on analyzing the rates of the optimization algorithms obtained with such high order discretization methods, but this is not the topic for this paper.
>
> ---

---

> > ### Author Response · Authors · 2021-11-15
> > **Response to Reviewer AvzH (Part 2)**
> >
> > **Problem 4**: *Suppose we consider second-order dynamic systems, can the finite-time convergence still hold in this case?*
> >
> > **Response 4**: Thank you for this interesting question, it is a topic we are exploring, currently we have no clear answer to it. Nesterov acceleration has been shown to correspond to an second-order ODE (Su et al., 2014), which motivates us to extend our research, we hope to have some affirmative answer in the future.
> >
> > ---
> >
> > References:
> >
> > [1] Karimi, Hamed, Julie Nutini, and Mark Schmidt. "Linear convergence of gradient and proximal-gradient methods under the polyak-łojasiewicz condition." Joint European Conference on Machine Learning and Knowledge Discovery in Databases. Springer, Cham, 2016.
> >
> > [2] Ghadimi, Saeed, Guanghui Lan, and Hongchao Zhang. "Mini-batch stochastic approximation methods for nonconvex stochastic composite optimization." Mathematical Programming 155.1-2 (2016): 267-305.
> >
> > [3] Fang, Cong, et al. "SPIDER: near-optimal non-convex optimization via stochastic path integrated differential estimator." Proceedings of the 32nd International Conference on Neural Information Processing Systems. 2018.
> >
> > [4] Su, Weijie, Stephen Boyd, and Emmanuel Candes. "A differential equation for modeling Nesterov’s accelerated gradient method: Theory and insights." Advances in neural information processing systems 27 (2014): 2510-2518.

---

### Official Review · Reviewer_rpnN · 2021-11-05

**Correctness:** 3
**Technical Novelty And Significance:** 3
**Empirical Novelty And Significance:** 2
**Recommendation:** 5
**Confidence:** 4

**Main Review:**

**Strength**
- Fundamental methods: This paper studies two (discrete) fundamental methods, which are gradient methods (with or without sign function) with gradient norm dependent step sizes. Many practical gradient methods in deep learning use such step sizes, so studying their fundamental versions are of interest.
- Experiments: Although only one very practical problem is considered, the experiments illustrate that the proposed methods outperform ADAM and Nesterov's accelerated GD.

**Weakness**
- Motivation: Although this work has merit, it is not well motivated. For example, it would have been nice if the authors provided references to the sentence "Concerning that many ~" in page 1. In specific, I have hoped to see some reasoning why we need to study such two specific gradient flows (and their discretization) over the standard gradient flow in Introduction and Related work sections. In addition, why is the gradient dominant condition of interest, over existing standard conditions, especially considering the authors' interest in applying their method to deep learning?
- Finite-time convergence in continuous time translates to acceleration in discretization: The authors mention that many existing related analyses focus on matching convergence rates from the continuous-time domain into the discrete-time domain. Instead, this paper states that (e.g., in page 8) finite-time convergence in continuous time translates to some acceleration in the associated discretized algorithm. I think matching the convergence rates as other papers did would have better convinced the readers on the theoretical justification of acceleration (in early iterations).
- Gradient dominant condition: There is no comparison to convergence rates of other methods under the same condition. Since the authors claim that their methods are faster than other methods in experiment, it would be nice to have some theoretical comparison.
- Experiment: Intuition behind the practical acceleration of the proposed methods is barely given. Therefore, I am not optimistic that this trend will appear in other experiments. What does the tuned parameter $q$ mean in the experiment on SVHN dataset?

**Minor**
- Page 3 below (6b): $F_d: Z_+ \times R^m \to R^m$
- Page 3 Examples 1 and 2: A feedback-based step size, component-wise step size and a backward Euler discretization are not used in the paper hereafter, so I think it is better to make this page concise, and state something more related to the main part.
- Page 6: Do the rates of the discretized methods match those of their continuous ones?
- Page 6: Could you explain further about the choice of $\ell_\infty$-norm, rather than $\ell_2$?
- Page 7 Example 1: For the case $q>1$ in example 1, it does not seem Lipschitz smooth given in Assumption 2. For an "academic" example, I think one would hope to see an example satisfying all of the considered conditions.
- Page 8 Figure 1 left: How did you define the optimality gap in the figure? (e.g., after certain number of iterations?)
- Page 9: How about the vanilla GD or the heavy-ball momentum? Using acronym GD (and SGD) for Nesterov's method seems odd.
- Page 9: How did you get 40 min here?
- Page 9 Figure 3: Why do we have drastic drops in the right figure?

**Summary Of The Paper:**

This paper studies the convergence rates of the two first-order methods, named $q$-RGF and $q$-SGF. These are constructed by forward Euler discretizing the $q$-rescaled gradient flow ($q$-RGF) [Wibisono et al., 2016] and $q$-signed GF ($q$-SGF) [Romero-Benosman, 2020], respectively. These gradient flows are shown to converge in finite time in [Romero-Benosman, 2020], under the gradient dominant condition of order $q$. The authors show that their forward Euler discretized versions have linear rates, under an additional Lipschitz smoothness of order $q$. The paper also considers their stochastic variants. Numerical experiments on toy examples and practical example illustrate that the proposed method might have practical advantage over existing methods such as GD and ADAM.

**Summary Of The Review:**

While the theoretical results seem new, they are not well motivated (at least practically), and the paper does not seem to provide sufficient justification of the success of the proposed methods in practical applications.

---

> ### Author Response · Authors · 2021-11-15
> **Response to Reviewer rpnN**
>
> Thank you very much for your time and we appreciate your review. Please see below our response to your review.
>
> ---
>
> **Problem 1**: *Although this work has merit, it is not well motivated. For example, it would have been nice if the authors provided references to the sentence "Concerning that many ~" in page 1. In specific, I have hoped to see some reasoning why we need to study such two specific gradient flows (and their discretization) over the standard gradient flow in Introduction and Related work sections. In addition, why is the gradient dominant condition of interest, over existing standard conditions, especially considering the authors' interest in applying their method to deep learning?*
>
> **Response 1**: Thank you very much for the question. First, the gradient flows that we consider in this paper are generalizations of existing standard gradient flow, e.g., classical gradient flow (which is in fact a 2-RGF), also the normalized gradient flow (which is in fact a $\infty$-RGF) and sign-based flow (which is in fact a $\infty$-SGF), proposed in Cortes (2006).
>
> As shown in Wibisono et al. (2016), we know that $q$-RGF is the continuous-time limit of higher-order gradient algorithm. Intuitively, with higher-order information we should expect a better convergence of the algorithm, with a trade-off of complexity in acquiring higher-order information, which is especially unfavorable in problems like deep learning. Therefore, a natural idea is that we focus on first-order discretization, which leads to the algorithms that we propose here.
>
> Also for the motivation of SGF, we want to further argue that recently there have been a lot of interest in Scaled SignGD based algorithms for distributed optimization,  which are closely related to our SGF, e.g., Karimireddy et al. (2019) and  Li et al. (2021).
>
> Regarding the generalized gradient dominance condition, it is a very important assumption in nonconvex optimization (of course there are many other works further relaxing the assumption or using other settings). We agree that it is not that weak, but we also want to emphasize that it is in fact widely verified in practice, e.g., in Mei et al. (2021), they showed that in reinforcement learning with softmax policy parameterization, the value function will satisfy \L{}ojasiewicz inequality (or KL inequality (Kurdyka, 1998)), which corresponds to the gradient dominance condition in our setting; also in deep learning there is evidence that gradient dominance does hold locally, e.g., Zhou and Liang, (2017). So we believe that our setting is still reasonable and useful.
>
> ---
>
> **Problem 2**: *The authors mention that many existing related analyses focus on matching convergence rates from the continuous-time domain into the discrete-time domain. Instead, this paper states that (e.g., in page 8) finite-time convergence in continuous time translates to some acceleration in the associated discretized algorithm. I think matching the convergence rates as other papers did would have better convinced the readers on the theoretical justification of acceleration (in early iterations).*
>
> **Response 2**: The sentence "Many of these papers also focus on explicit mapping and matching of convergence rates from the continuous-time domain into the discrete-time domain" is misleading, and has been rephrased in the revised version. What we intended was to emphasize the new trend of works trying to connect dynamical system flows with optimization algorithms, similarly to what we are doing here. The goal is to propose new flows based on dynamical systems theory and then try to analyze their convergence rate in discrete time. The fact is, there are no guarantees that explicit discretization, like forward Euler, would lead to any direct convergence rate matching. However, this should not stop us from exploring new flows and their associated novel algorithms, in search for better rates and better empirical performances. The paper has been modified to clarify this point.
>
> ---

---

> > ### Author Response · Authors · 2021-11-15
> > **Response to Reviewer rpnN (Part 2)**
> >
> > **Problem 3**: *There is no comparison to convergence rates of other methods under the same condition. Since the authors claim that their methods are faster than other methods in experiment, it would be nice to have some theoretical comparison*
> >
> > **Response 3**: Thank you very much for the suggestion. In fact besides the comparison viewpoint, our proposed discrete-time algorithms can work in a very general setting, because we extend the standard smoothness and gradient dominance assumptions to more generalized conditions (as we mentioned in Remark 3). Under weaker assumptions, there should be more optimization problems that the proposed algorithms can efficiently solve, comparatively to standard algorithms, which translates to the observed experiments results.
> >
> > Regarding the theoretical comparison between rates, there may not be readily available because the setting is different. However, as mentioned in Remark 3, the proposed algorithms reduce or recover classical results when we set $p=2$, so we believe that recovering exiting rates, in specific settings, can also be regarded as a theoretical comparison with some existing rates.
> >
> > ---
> >
> > **Problem 4**: *Intuition behind the practical acceleration of the proposed methods is barely given. Therefore, I am not optimistic that this trend will appear in other experiments. What does the tuned parameter  mean in the experiment on SVHN dataset?*
> >
> > **Response 4**: We agree that at this stage there is no constructive precise way to tune the parameter $q$. We know that it has to be larger than the order of gradient dominance $p$, however, the order $p$ is not explicit in most general non-convex problems. For instance, in DNN we have some indications that some DNNs (and loss functions) are gradient dominant of order $2$, e.g., Zhou and Liang, (2017). This allows us to choose a $q$ that is larger than $2$, we then start increasing $q$ until we see a deterioration in the convergence performance. We agree that this is not an ideal nor an intuitive tuning, but in all fairness, we hardly see ideal or intuitive tuning being applied to most state of the art optimization algorithms for DNN applications, where hyper-parameters are tuned empirically by trial and error.
> >
> > ---
> >
> > **Other Problems**
> >
> > ---
> >
> > **Problem**: *Page 3 below (6b): $F_d: Z_+\times \mathbb{R}^m\rightarrow\mathbb{R}^m$*
> >
> > **Response**: We corrected the typo, thank you very much.
> >
> > ---
> >
> > **Problem**: *Page 3 Examples 1 and 2: A feedback-based step size, component-wise step size and a backward Euler discretization are not used in the paper hereafter, so I think it is better to make this page concise, and state something more related to the main part.*
> >
> > **Response**: We agree, these examples were meant to clarify the general formulation in (6), so confusing details about feedback-based step size, etc. are unnecessary and have been removed in the revised version of the paper.
> >
> > ---
> >
> > **Problem**: *Page 6: Do the rates of the discretized methods match those of their continuous ones?*
> >
> > **Response**: Thank you for the question. We do not think these two rates should be compared. Indeed, in Theorem 1 the finite-time convergence result in continuous-time guarantees that the state $x(t)$ reaches the equilibrium point $x^*$, and stays at this point (by definition of equilibrium point $\frac{d x(t)}{dt}|_{t=t^*}=0$; see Definition 2 in Appendix B of the paper). However, the concept of finite-time convergence in continuous-time does not translate to a similar concept of finite-time convergence in discrete-time. In the sense that we were not expecting to be able to converge in a desired predefined number of discrete steps. All what we are hoping for is to derive some competitive convergence rates after discretizing the continuous flow, and to observe empirical acceleration w.r.t. some existing methods.
> >
> > ---

---

> > > ### Author Response · Authors · 2021-11-15
> > > **Response to Reviewer rpnN (Part 3)**
> > >
> > >
> > >
> > > **Problem**: *Page 6: Could you explain further about the choice of $\ell_\infty$-norm, rather than $\ell_2$?*
> > >
> > > **Response**: Thank you for the question. Here $\ell_\infty$-norm is applied in the SGF analysis. Besides the fact that many papers on sign-based algorithms have applied it (as we mentioned in Remark 4), maybe we can further argue it justification from a more theoretical viewpoint.
> > >
> > > Here the norm equipped with the gradient, as defined in SGF, is the $\ell_1$-norm, so we can say that the dual space (the space of gradients $\nabla f(x)$) is equipped with $\ell_1$-norm; so its counterpart, the primal space (the space of the variable $x$) should be equipped with the dual norm of $\ell_1$, i.e., $\ell_\infty$-norm. So the choice of $\ell_\infty$-norm in the analysis is natural. Generally in the $\ell_2$ case we do not mention the difference because the dual norm of $\ell_2$-norm is just itself, so basically the primal and dual spaces are identical and we will not distinguish them; but here with a different norm, we need to specify the $(\ell_1,\ell_\infty)$-norm pair.
> > >
> > > In practice, of course the gradient can be measured under many norms ($\ell_1$, $\ell_2$ or $\ell_\infty$...). Therefore, we have that the Lipschitz (or Holder) smoothness parameter will be different under different norms, e.g., if a function is $L$-Lipschitz smooth under $\ell_2$-norm, then it will be "at most" $dL$-Lipschitz smooth under $\ell_\infty$-norm (the actual parameter can be smaller) because $\|x\|_2^2\leq d\|x\|_\infty^2$ where $d$ is the dimension number. Also there are some works discussing the smoothness parameters under different norms, e.g., Balles et al. (2020), while here we will not discuss it in details.
> > >
> > > ---
> > >
> > > **Problem**: *Page 7 Example 1: For the case $q>1$ in example 1, it does not seem Lipschitz smooth given in Assumption 2. For an "academic" example, I think one would hope to see an example satisfying all of the considered conditions.*
> > >
> > > **Response**: Thank you for the point, and sorry for the confusion. In fact as mentioned in Lei and Ying, (2020), our example is a special case of the so-called $q$-th power absolute distance loss, and it will be Lipschitz smooth with of order $q$ when $q\in (0,1]$, so the proposed example in fact satisfy the requirement in this regime, we have revised the description of this part in the submission and updated the experiment result with more information (following another reviewer's suggestion).
> > >
> > > ---
> > >
> > > **Problem**: *Page 8 Figure 1 left: How did you define the optimality gap in the figure? (e.g., after certain number of iterations?)*
> > >
> > > **Response**: Here the optimality gap is the common criteria $f(x_t)-f^*$ where $f^*$ is the optimal point and $x_t$ is the last iterate variable. For each setting, we run it for 1000 iterations and get the optimality gap value. Also for this example we know that the optimal value is $f^*=f(0)=0$.
> > >
> > > ---
> > >
> > > **Problem**: *Page 9: How about the vanilla GD or the heavy-ball momentum? Using acronym GD (and SGD) for Nesterov's method seems odd.*
> > >
> > > **Response**: We apologize for this typo. Indeed, the tests reported in this paper are about Euler discretization of q-RGF, and q-SGF, and its comparison to classical GD. No Nesterov acceleration has been used. We have tested Nesterov accelerated gradient and other momentum-based methods, against different higher order discretization of q-RGF, and q-SGF. However, since we are still working on deriving proper theoretical convergence rates for these higher order discretization, we are not reporting these comparisons in this paper. We have corrected this typo in the revised version of the paper.
> > >
> > > ---
> > >
> > > **Problem**: *Page 9: How did you get 40 min here?*
> > >
> > > **Response**: We report in Figure 3 the training loss as function of the training iterations, where we see a lead of q-RGF/SGF in the early iterations. However, to make sure that this lead translates into real computation time lead, we plotted, in Figure 4,  the same training loss as function of the computation time, where we see an average lead of about 2500 sec (about 40 mins).
> > >
> > > ---

---

> > > > ### Author Response · Authors · 2021-11-15
> > > > **Response to Reviewer rpnN (Part 4)**
> > > >
> > > > **Problem**: *Page 9 Figure 3: Why do we have drastic drops in the right figure?*
> > > >
> > > > **Response**: It is due to the way we implemented the tests. Indeed, we did the tests on the test-set in parallel with the training. In other words, we run the model on the test set after each training epoch. This was done to be able to monitor that the model is not overfitting the training data (in which case the test plot would be going up). This explains the observed jump in test performance after a few iterations.
> > > >
> > > > References:
> > > >
> > > > [1] Cortés, Jorge. "Finite-time convergent gradient flows with applications to network consensus." Automatica 42.11 (2006): 1993-2000.
> > > >
> > > > [2] Wibisono, Andre, Ashia C. Wilson, and Michael I. Jordan. "A variational perspective on accelerated methods in optimization." proceedings of the National Academy of Sciences 113.47 (2016): E7351-E7358.
> > > >
> > > > [3] Karimireddy, Sai Praneeth, et al. "Error feedback fixes signsgd and other gradient compression schemes." International Conference on Machine Learning. PMLR, 2019.
> > > >
> > > > [4] Li, Xiuxian, et al. "On Faster Convergence of Scaled Sign Gradient Descent." arXiv preprint arXiv:2109.01806 (2021).
> > > >
> > > > [5] Mei, Jincheng, et al. "Leveraging non-uniformity in first-order non-convex optimization." arXiv preprint arXiv:2105.06072 (2021).
> > > >
> > > > [6] Balles, Lukas, Fabian Pedregosa, and Nicolas Le Roux. "The Geometry of Sign Gradient Descent." arXiv preprint arXiv:2002.08056 (2020).
> > > >
> > > > [7] Zhou, Yi, and Yingbin Liang. "Characterization of gradient dominance and regularity conditions for neural networks." arXiv preprint arXiv:1710.06910 (2017).
> > > >
> > > > [8] Kurdyka, Krzysztof. "On gradients of functions definable in o-minimal structures." Annales de l'institut Fourier. Vol. 48. No. 3. 1998.
> > > >
> > > > [9] Lei, Yunwen, and Yiming Ying. "Fine-grained analysis of stability and generalization for stochastic gradient descent." International Conference on Machine Learning. PMLR, 2020.

---

> ### Comment · Reviewer_rpnN · 2021-11-27
> **Thanks for the detailed response**
>
> and thanks for your patience.
>
> - Relation to higher-order method: I agree that (first-order) discretizing the continuous-time limit of higher-order gradient algorithm has some potential to yield a fast rate, and can be of interest. However, I do not see any theoretical evidence in the paper whether this approach has merit from resembling higher-order method, over existing methods. So, I am not fully convinced with the authors' claim.
> - Holder smoothness: As remark 3 points out, one of the main contribution of this work is weakening the Lipschitz smoothness assumption for the RGF and SGF to Holder smoothness. Existing works assume Lipschitz smoothness and gradient dominance. I thus agree with reviewer xRJ7 that this may not be a significant contribution. I suggest explicitly mentioning this in the beginning of the paper. This was not clear before remark 3 in page 6.
> - Finite-time convergence and discrete method:  To be honest, I am not yet fully convinced with the authors' claim relating the finite-time convergence and their convergence results of discrete method. As I said before, like other existing literature, matching the rates would have been more convincing and interesting in my viewpoint.
> - Example 1: Explanations in Example 1 still seem a bit confusing and redundant.
> - Figure 1: I suggest explicitly stating that you compare optimality gap after 1000 iterations.
> - Figure 3: Is testing on the test-set in parallel with the training standard? Do you at least conjecture that the trend will be the same for the training without testing on the test-set in parallel? I suggest explicitly stating the details of the experiment at least in the Appendix.
>
> I appreciate the authors' hard work, but after reading the authors' response and reviewers' comments,  I decided to maintain my score.

---

> > ### Author Response · Authors · 2021-11-29
> > **Thank you and Further Response**
> >
> > Thank you for the response. We want to further elaborate on our rebuttal to defend our paper:
> >
> > **"Relation to higher-order method" (motivation of RGF)**:
> >
> > First we are not totally clear about what is the "merit" that you are aiming for, maybe some rates better than the current linear rate (in Theorem 3)? But first-order methods has known fundamental computation limits (or lower bound) for solving such problems, existing works on PL conditions also have a linear rate (and by the way weaker than ours!), so currently it is hard to expect to have a superlinear convergence rate. Maybe a better rate can be attained under stronger assumptions like higher-order smoothness (which often appears in higher-order method literature), but it is out of scope of this work because we only assume-easier to satisfy- first-order conditions.
> >
> > Second we want to point out that, as mentioned before, classical gradient flow and normalized gradient flow can be incorporated into the RGF framework. Classical gradient flow is the most basic one, while normalized gradient flow is known for stabilizing the training, and it is restrictive in the sense that it only uses the direction information. As an intermediate solution, it is interesting to study the generalized framework of RGF, which is what we are proposing here.
> >
> > **"Holder smoothness (and generalized gradient dominance)"**:
> >
> > First we want to point out that the gradient dominance here is also a generalization of existing results. Similar to our reply to Reviewer xRJ7, the generalized gradient dominance condition will be satisfied in several important examples like reinforcement learning (Mei et al. (2021)). Furthermore, the extended smoothness and gradient dominance conditions also brought non-trivial challenges in the analysis part, and lead to better-linear- rates, which we already mentioned in the reply to Reviewer xRJ7.
> >
> > We also want to point out that our contributions in this work come from both continuous and discrete time aspects, so we respectfully disagree with you in the sense that we believe that our extension is a nontrivial and leads to more general analysis, compared to existing work.
> >
> > **"Finite-time convergence and discrete method"**:
> >
> > Here we respectfully disagree with your point. As we replied in *"Problem: Page 6: Do the rates of the discretized methods match those of their continuous ones?"*, our finite-time convergence stems from **finite-time stability** argument, which is an important topic in control community. Here we do NOT try to derive some weaker complexity results like $O(\text{poly}(1/t))$ rate for the continuous-time part (if we understand your "matching the rates" correctly). So we do not think these two results should be compared;  there should NOT be a matching issue (there is no similar finite-time meaning in discrete-time, as we explained before), and we believe it should NOT be considered to be weakness in our work.
> >
> > For the numerical tests, we have clarified Example 1 and further show that we obtain numerically exactly the optimal step-size predicted by our theory! The DNN tests follow standard testing procedures, where the network is trained on a training set and tested on a DIFFERENT testing set. Since we cannot re-upload a revision at this stage, we will add more explanations about the numerical testing procedure in the final version of the paper, thank you for your suggestions.
> >
> > We truly appreciate your opinion about our work, we hope our explanations can help to alleviate any remaining misunderstandings and concerns. With the above additional explanations, we definitely hope that you can re-evaluate your decision. Thank you.

---

> > > ### Comment · Reviewer_rpnN · 2021-11-30
> > > **Thanks for your fast and detailed response.**
> > >
> > > - **Relation to higher-order method" (motivation of RGF)**: My reply was in response to your statement about higher-order information: "As shown in Wibisono et al. (2016), we know that q-RGF is the continuous-time limit of higher-order gradient algorithm. Intuitively, with higher-order information we should expect a better convergence of the algorithm, with a trade-off of complexity in acquiring higher-order information, which is especially unfavorable in problems like deep learning. Therefore, a natural idea is that we focus on first-order discretization, which leads to the algorithms that we propose here." I just was not able to fully agree with such statement without any theoretical or practical justification, although I knew why you mentioned it. This is not mentioned in the paper and is not that important, so further discussion does not seem necessary.
> > > - **Holder smoothness (and generalized gradient dominance)**: Remark 3 seems to say that the only difference between this paper and (Karimi et al, 2016, Beznosikov et al, 2020) are the smoothness. I now forgot the very details of the paper, but if you are claiming that the assumption on the gradient dominance is also generalized, I suggest modifying remark 3. I also suggest to explicitly state the difference of this paper to (Karimi et al, 2016, Beznosikov et al, 2020) from the beginning, since the introduction (especially the contribution section) does not specify this specific contribution clearly. Also, I initially thought that the first-order discretization of q-RGF (2) and q-SGF(3) is new in this paper, and I think the introduction can be improved.
> > > - **Finite-time convergence and discrete method**: I totally understand the authors' side. I think this discrepancy stems from our different viewpoints. First, my evaluation mainly focused on the discrete-time analysis, since the discrete version is the one that we use in practice. In terms of this discrete-time aspect, it remains one to evaluate whether this is significant or not. I personally thought such extension not significant enough for publication in ICLR, but I agree that this is debatable. I also value the continuous-time analysis, but I was not fully convinced that this specific continuous-time analysis gives a new relevant(?) insight on the discrete-time analysis. This part is the one that seems to have a big difference between myself and the authors. I really appreciate hard works by authors (also including its stochastic extension and experiments), but as a reviewer, I have to make a decision on papers and I think this paper stays on a borderline.

---

> > > > ### Author Response · Authors · 2021-11-30
> > > > **Thank you**
> > > >
> > > > We really appreciate your opinions in our work, even though there is still some disagreement between us on the contributions in our work, which may be hard to solve at this time. We are very glad that our message seems to be delivered clearly and that we engaged in this discussion. We will make further revisions in the final version following your suggestions. Thank you very much for the response.

---

### Author Response · Authors · 2021-11-15
**Summary of Revised Submission**

We appreciate all reviewers for the constructive and detailed feedback. Following reviewers' suggestions, we have revised our submission and uploaded it here. The main changes have been highlighted in blue. Below is a summary of the main changes:
- We added more discussion on the motivation of the gradient flow and main assumptions.
- Following the suggestion of Reviewer 5qiS, we revised the numerical experiment.
- Some reviewers have concerns on the batch size in the stochastic setting, we have added discussion in Remark 6 to explain the results.
- Following the suggestions of reviewers, we corrected some typos and rephrased some sentences to improve the presentation.

---

> ### Author Response · Authors · 2021-11-22
> **Thank you to the reviewers' effort and time; did our replies answer your main concerns ?**
>
> Dear Reviewers,
> We appreciate your work and time put into the review of our paper. We also appreciate that you are busy (we are all in the same boat of being authors and reviewers, etc.). We have spent a lot of time in reading and discussing your concerns among ourselves, and then much time in revising the paper and drafting what we believe are thorough responses to each reviewer.
> We would really appreciate if you could read our replies and let us know if your main concerns have been addressed, or if you have anymore questions for us. Alternatively, if you found our responses to be satisfactory, we would appreciate if you could revise your scores accordingly.
> Thank you again for your work.
> Best wishes,
> Authors

---

### Decision · Program_Chairs · 2022-01-20

**Decision:**

Reject

**Comment:**

This paper starts from the observation that a certain class of rescaled gradient flows - referred to in the paper as RGF and SGF - converge to a solution in finite time (Wibisono et al., 2016; Romero and Benosman, 2020). As a result, it is plausible to ask whether the Euler discretizations of these flows - viewed now as optimization algorithms - enjoy superior convergence properties or not. The authors' main results establish a linear convergence rate under a certain gradient dominance condition, as well as linear convergence to an $\epsilon$-neighborhood of a solution if the algorithms are run with minibatch gradients of size $O(1/\epsilon^\rho)$ for some positive exponent $\rho>0$.

The reviewers raised several concerns regarding the motivation of the authors' work and the comparison of the rates they obtain to other related papers in the literature. The reviewers that raised these concerns were not convinced by the authors' rebuttal and maintained their original assessment during the discussion phase.

From my own reading of the paper, I was perplexed by the fact that the authors did not compare the rates they obtained to existing results in the context of KL optimization, such as the cited paper by Attouch and Bolte and many follow-up works in the area. Also, in the stochastic part, while the authors argue that "utilizing batches with size dependent on $1/\epsilon$ is absolutely reasonable and usual, in both theory and practice", it should be noted that a high accuracy requirement (small $\epsilon$) could lead to completely unreasonable batch sizes (effectively exceeding the size of the dataset, especially when $\psi$ is small). Thus, while it is possible to achieve convergence to arbitrarily high accuracy with a sufficiently small step-size for a _fixed_ batch size, the rate of this convergence cannot be linear overall - in contrast to the way that the authors frame their result.

In view of the above, I concur that the paper does not clear the bar for ICLR, so I am recommending rejection at this stage (but I would encourage the authors to resubmit a suitably revised version of their paper at the next opportunity).